# Open-YOLO 3D: Towards Fast and Accurate Open-Vocabulary 3D Instance Segmentation

**Mohamed El Amine Boudjoghra**
TUM, MBZUAI

**Angela Dai**
TUM

**Jean Lahoud**
MBZUAI

**Hisham Cholakkal**
MBZUAI

**Rao Muhammad Anwer**
MBZUAI, Aalto University

**Salman Khan**
MBZUAI, ANU

**Fahad Shahbaz Khan**
MBZUAI, Linköping University

## Abstract

Recent works on open-vocabulary 3D instance segmentation show strong promise but at the cost of slow inference speed and high computation requirements. This high computation cost is typically due to their heavy reliance on aggregated clip features from multi-view, which require computationally expensive 2D foundation models like Segment Anything (SAM) and CLIP. Consequently, this hampers their applicability in many real-world applications that require both fast and accurate predictions. To this end, we propose a novel open-vocabulary 3D instance segmentation approach, named Open-YOLO 3D, that efficiently leverages only 2D object detection from multi-view RGB images for open-vocabulary 3D instance segmentation. We demonstrate that our proposed Multi-View Prompt Distribution (MVPDist) method makes use of multi-view information to account for misclassification from the object detector to predict a reliable label for 3D instance masks. Furthermore, since projections of 3D object instances are already contained within the 2D bounding boxes, we show that our proposed low granularity label maps, which require only a 2D object detector to construct, are sufficient and very fast to predict prompt IDs for 3D instance masks when used with our proposed MVPDist. We validate our Open-YOLO 3D on two benchmarks, ScanNet200 and Replica, under two scenarios: *(i)* with ground truth masks, where labels are required for given object proposals, and *(ii)* with class-agnostic 3D proposals generated from a 3D proposal network. Our Open-YOLO 3D achieves state-of-the-art performance on both datasets while obtaining up to ∼16× speedup compared to the best existing method in literature. On ScanNet200 val. set, our Open-YOLO 3D achieves mean average precision (mAP) of 24.7% while operating at 22 seconds per scene. github.com/aminebdj/OpenYOLO3D

## 1 Introduction

3D instance segmentation is a computer vision task that involves the prediction of masks for individual objects in a 3D point cloud scene. It holds significant importance in fields like robotics and augmented reality. Due to its diverse applications, this task has garnered increasing attention in recent years. Researchers have long focused on methods that typically operate within a closed-set framework, limiting their ability to recognize objects not present in the training data. This constraint poses challenges, particularly when novel objects must be identified or categorized in unfamiliar environments. Recent methods Nguyen et al. (2024); Takmaz et al. (2023) address the problem of novel class segmentation, but they suffer from slow inference that ranges from 5 minutes for small scenes to 10 minutes for large scenes due to their reliance on computationally heavy foundation models like SAM Kirillov et al. (2023) and CLIP Zhang et al. (2023) along with heavy computation for lifting 2D CLIP feature to 3D.

Open-vocabulary 3D instance segmentation robotics tasks such as manipulating objects and inventory management require accurate predictions while being fast in the decision-making process. Furthermore, these tasks alter the point cloud throughout time, where objects can be rearranged, removed, or added; this would require open vocabulary 3D instance segmentation pipelines to re-run from

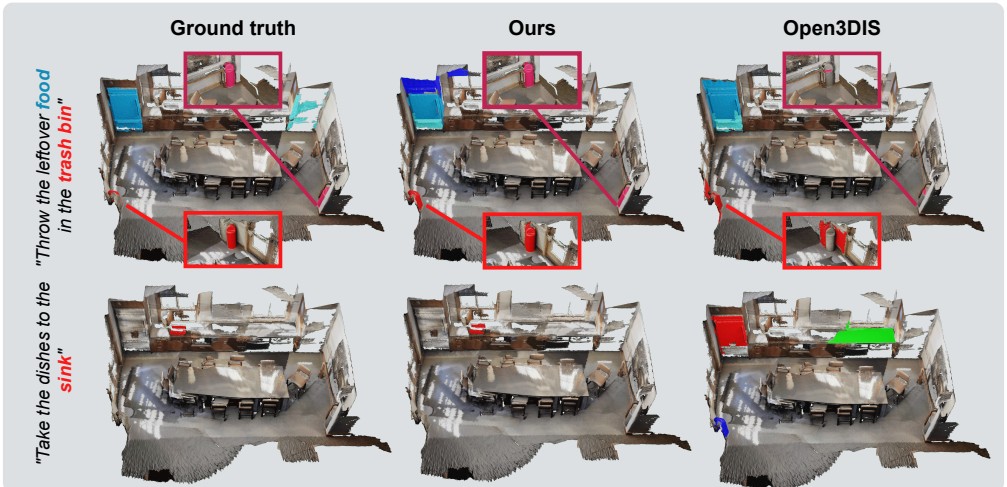

Figure 1: **Open-vocabulary 3D instance segmentation with our Open-YOLO 3D.** The proposed Open-YOLO 3D is capable of segmenting objects in a zero-shot manner. Here, We show the output for a ScanNet200 Rozenberszki et al. (2022) scene with various prompts, where our model yields improved performance compared to the recent Open3DIS Nguyen et al. (2024). We show zoomed-in images of hidden predicted instances in the colored boxes. Additional results are in suppl. material.

scratch for every updated scene. Thus, recent methods would be ill-suited for such robotics tasks due to their low speed. Motivated by recent advances in 2D object detection Cheng et al. (2024), we look into an alternative approach that efficiently leverages fast object detectors instead of utilizing computationally expensive foundation models that are adopted by recent methods.

This paper proposes a novel open-vocabulary 3D instance segmentation method, named Open-YOLO 3D, that utilizes efficient, joint 2D-3D information using bounding boxes and projections from 3D point clouds. We employ an open-vocabulary 2D object detector to generate bounding boxes with their class labels for all frames corresponding to the 3D scene; on the other side, we utilize a 3D instance segmentation network to generate 3D class-agnostic instance masks for the point clouds, which proves to be much faster than 3D proposal generation methods from 2D instances Nguyen et al. (2024); Lu et al. (2023). Unlike recent methods Takmaz et al. (2023); Nguyen et al. (2024) which use SAM and CLIP to lift 2D clip features to 3D for prompting the 3D mask proposal, we propose an approach that relies on the bounding box predictions from 2D object detectors which prove to be significantly faster. We use the predicted bounding boxes in all RGB frames corresponding to the point cloud scene to construct a Low Granularity (LG) label map for every frame. One LG label map is a two-dimensional array with the same height and width as the RGB frame, with the bounding box areas replaced by their predicted class label. Next, we use our proposed MVPDist to assign the best possible prompt ID to the 3D masks by using multi-view information, we present an example output of our method in Figure 1. Our contributions are the following:

- We introduce a 2D object detection-based approach for open-vocabulary labeling of 3D instances, which efficiently uses object detectors to greatly improve the results.
- We propose a novel approach to scoring 3D mask proposals using only bounding boxes from 2D object detectors.
- Our Open-YOLO 3D achieves superior performance on two benchmarks, while being considerably faster than existing methods in the literature. On ScanNet200 val. set, our Open-YOLO 3D achieves an absolute gain of 2.3% at mAP50 while being ∼16x faster compared to the recent Open3DIS Nguyen et al. (2024).

## 2 RELATED WORKS

**Closed-vocabulary 3D segmentation:** The 3D instance segmentation task aims at predicting masks for individual objects in a 3D scene, along with a class label belonging to the set of known classes.

Some methods use a grouping-based approach in a bottom-up manner, by learning embeddings in the latent space to facilitate clustering of object points Chen et al. (2021); Han et al. (2020); He et al. (2021); Jiang et al. (2020); Lahoud et al. (2019); Liang et al. (2021); Wang et al. (2018); Zhang & Wonka (2021). Conversely, proposal-based methods adopt a top-down strategy, initially detecting 3D bounding boxes and then segmenting the object region within each box Engelmann et al. (2020); Hou et al. (2019); Liu et al. (2020); Yang et al. (2019); Yi et al. (2019). Notably, inspired by advancements in 2D works Cheng et al. (2022; 2021), transformer designs Vaswani et al. (2017) have been recently applied to 3D instance segmentation tasks Schult et al. (2023); Sun et al. (2023); Kolodiazhnyi et al. (2024); Al Khatib et al. (2023); Jain et al. (2024). Mask3D Schult et al. (2023) introduces the first hybrid architecture that combines Convolutional Neural Networks (CNN) and transformers for this task. It uses a 3D CNN backbone to extract per-point features and a transformer-based instance mask decoder to refine a set of queries. Building on Mask3D, the authors of Al Khatib et al. (2023) show that using explicit spatial and semantic supervision at the level of the 3D backbone further improves the instance segmentation results. Oneformer3D Kolodiazhnyi et al. (2024) follows a similar architecture and introduces learnable kernels in the transformer decoder for a unified semantic, instance, and panoptic segmentation. ODIN Jain et al. (2024) proposes an architecture that uses 2D-3D fusion to generate the masks and class labels. Other methods introduce weakly-supervised alternatives to dense annotation approaches, aiming to reduce the annotation cost associated with 3D data Chibane et al. (2022); Hou et al. (2021); Xie et al. (2020). While these methodologies strive to enhance the quality of 3D instance segmentation, they typically rely on a predefined set of semantic labels. In contrast, our proposed approach aims at segmenting objects with both known and unknown class labels.

**Open-vocabulary 2D recognition:** This task aims at identifying both known and novel classes, where the labels of the known classes are available in the training set, while the novel classes are not encountered during training. In the direction of open-vocabulary object detection (OVOD), several approaches have been proposed Zhong et al. (2022); Pham et al. (2024); Liu et al. (2023); Zang et al. (2022); Wang et al. (2023); Kaul et al. (2023); Yao et al. (2023); Cheng et al. (2024). Another widely studied task is open-vocabulary segmentation (OVSS) Bucher et al. (2019); Xu et al. (2022); Li et al. (2021); Ghiasi et al. (2022); Liang et al. (2023). Recent open-vocabulary semantic segmentation methods Li et al. (2021); Ghiasi et al. (2022); Liang et al. (2023) leverage pre-trained CLIP Zhang et al. (2023) to perform open-vocabulary segmentation, where the model is trained to output a pixel-wise feature that is aligned with the text embedding in the CLIP space. Furthermore, AttrSeg Ma et al. (2024) proposes a decomposition-aggregation framework where vanilla class names are first decomposed into various attribute descriptions, and then different attribute representations are aggregated into a final class representation. Open-vocabulary instance segmentation (OVIS) aims at predicting instance masks while preserving high zero-shot capabilities. One approach Huynh et al. (2022) proposes a cross-modal pseudo-labeling framework, where a student model is supervised with pseudo-labels for the novel classes from a teacher model. Another approach VS et al. (2023) proposes an annotation-free method where a pre-trained vision-language model is used to produce annotations at both the box and pixel levels. Although these methods show high zero-shot performance and real-time speed, they are still limited to 2D applications only.

**Open-vocabulary 3D segmentation:** Several methods Huang et al. (2024); Peng et al. (2023); Gu et al. (2023); Hong et al. (2023) have been proposed to address the challenges of open-vocabulary semantic segmentation where they use foundation models like clip for unknown class discovery, while the authors of Boudjoghra et al. (2023) focus on weak supervision for unknown class discovery without relying on any 2D foundation model. OpenScene Peng et al. (2023) makes use of 2D open-vocabulary semantic segmentation models to lift the pixel-wise 2D CLIP features into the 3D space, which allows the 3D model to perform 3D open-vocabulary point cloud semantic segmentation. On the other hand, ConceptGraphs Gu et al. (2023) relies on creating an open-vocabulary scene graph that captures object properties such as spatial location, enabling a wide range of downstream tasks including segmentation, object grounding, navigation, manipulation, localization, and remapping. In the direction of 3D point cloud instance segmentation, OpenMask3D Takmaz et al. (2023) uses a 3D instance segmentation network to generate class-agnostic mask proposals, along with SAM Kirillov et al. (2023) and CLIP Zhang et al. (2023), to construct a 3D clip feature for each mask using RGB-D images associated with the 3D scene. Unlike OpenMask3D where a 3D proposal network is used, OVIR-3D Lu et al. (2023) generates 3D proposals by fusing 2D masks obtained by a 2D instance segmentation model. Open3DIS Nguyen et al. (2024) combines proposals from 2D and 3D with novel 2D masks fusion approaches via hierarchical agglomerative clustering, and also proposes to

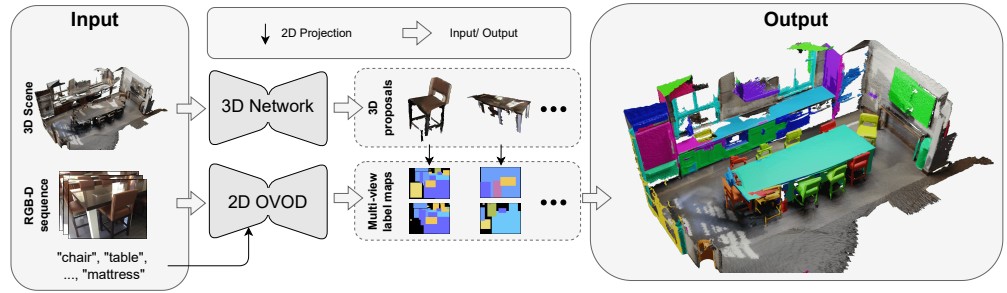

Figure 2: **Proposed open-world 3D instance segmentation pipeline.** We use a 3D instance segmentation network (3D Network) for generating class-agnostic proposals. For open-vocabulary prediction, a 2D Open-Vocabulary Object Detector (2D OVOD) generates bounding boxes with class labels. These predictions are used to construct label maps for all input frames. Next, we assign the top-k label maps to each 3D proposal based on visibility. Finally, we generate a Multi-View Prompt Distribution from the 2D projections of the proposals to match a text prompt to every 3D proposal.

use point-wise 3D CLIP features instead of mask-wise features. The two most recent approaches in Nguyen et al. (2024); Takmaz et al. (2023) show promising generalizability in terms of novel class discovery Takmaz et al. (2023) and novel object geometries especially small objects Nguyen et al. (2024). However, they both suffer from slow inference speed, as they rely on SAM for 3D mask proposal clip feature aggregation in the case of OpenMask3D Takmaz et al. (2023), and for novel 3D proposal masks generation from 2D masks Nguyen et al. (2024).

## 3 PRELIMINARIES

**Problem formulation:** 3D instance segmentation aims at segmenting individual objects within a 3D scene and assigning one class label to each segmented object. In the open-vocabulary (OV) setting, the class label can belong to previously known classes in the training set as well as new class labels. To this end, let $P$ denote a 3D reconstructed point cloud scene, where a sequence of RGB-D images was used for the reconstruction. We denote the RGB image frames as $\mathcal{I}$ along with their corresponding depth frames D. Similar to recent methods Peng et al. (2023); Takmaz et al. (2023); Nguyen et al. (2024), we assume that the poses and camera parameters are available for the input 3D scene.

### 3.1 BASELINE OPEN-VOCABULARY 3D INSTANCE SEGMENTATION

We base our approach on OpenMask3D Takmaz et al. (2023), which is the first method that performs open-vocabulary 3D instance segmentation in a zero-shot manner. OpenMask3D has two main modules: a class-agnostic mask proposal head, and a mask-feature computation module. The class-agnostic mask proposal head uses a transformer-based pre-trained 3D instance segmentation model Schult et al. (2023) to predict a binary mask for each object in the point cloud. The mask-feature computation module first generates 2D segmentation masks by projecting 3D masks into views in which the 3D instances are highly visible, and refines them using the SAM Kirillov et al. (2023) model. A pre-trained CLIP vision-language model Zhang et al. (2023) is then used to generate image embeddings for the 2D segmentation masks. The embeddings are then aggregated across all the 2D frames to generate a 3D mask-feature representation.

**Limitations**: OpenMask3D makes use of the advancements in 2D segmentation (SAM) and vision-language models (CLIP) to generate and aggregate 2D feature representations, enabling the querying of instances according to open-vocabulary concepts. However, this approach suffers from a high computation burden leading to slow inference times, with a processing time of 5-10 minutes per scene. The computation burden mainly originates from two sub-tasks: the 2D segmentation of the large number of objects from the various 2D views, and the 3D feature aggregation based on the object visibility. We next introduce our proposed method which aims at reducing the computation burden and improving the task accuracy.

## 4 METHOD: OPEN-YOLO 3D

**Motivation**: We here present our proposed 3D open-vocabulary instance segmentation method, Open-YOLO 3D, which aims at generating 3D instance predictions in an efficient strategy. Our proposed method introduces efficient and improved modules at the task level as well as the data level. *Task Level:* Unlike OpenMask3D, which generates segmentations of the projected 3D masks, we pursue a more efficient approach by relying on 2D object detection. Since the end target is to generate labels for the 3D masks, the increased computation from the 2D segmentation task is not necessary. *Data Level:* OpenMask3D computes the 3D mask visibility in 2D frames by iteratively counting visible points for each mask across all frames. This approach is time-consuming, and we propose an alternative approach to compute the 3D mask visibility within all frames at once.

### 4.1 OVERALL ARCHITECTURE

Our proposed pipeline is shown in Figure 2. First, we generate a set of instance proposals $M$ using a 3D instance segmentation network; the proposals are represented as binary masks, where every 3D mask has a dimension equal to the number of points as the input point cloud. For the open vocabulary prediction, we use a 2D open-vocabulary object detection model to generate a set of bounding boxes denoted $\mathbf{B}_i$ for every frame $\mathcal{I}_i$; the bounding boxes with their predicted labels are used to construct a low-granularity label map $\mathcal{L}_i$ for every input frame $\mathcal{I}_i$. To assign a prompt ID to the 3D mask proposals, we first start by projecting all $N$ points in the point cloud scene $P$ onto the $N_f$ frames, which results in $N_f$ 2D projection with $N$ points for each. Afterward, the 2D projections and the 3D mask proposals are used to compute the visibility of every mask in every frame using our proposed accelerated visibility computation (**VAcc**); the visibility is then used to assign top-k Low-Granularity label maps to each mask and to select the top-k 2D projections corresponding to every 3D mask proposal; for a single 3D mask we crop the (x, y) coordinates from the projections using the instance mask and filter out the points that are occluded or outside the frame. The final cropped (x, y) coordinates from the top-k frames are used to select per-point labels from their corresponding Low-Granularity label maps to finally construct a Multi-View Prompt Distribution to predict the prompt ID corresponding to the 3D mask proposal.

### 4.2 3D OBJECT PROPOSAL

To generate class-agnostic 3D object proposals, we rely on the 3D instance generation approach Mask3D Schult et al. (2023), which allows for faster proposal generation compared to 2D mask-based 3D proposal generation methods Nguyen et al. (2024); Lu et al. (2023). Mask3D is a hybrid model that combines a 3D Convolutional Neural Network as a backbone for feature generation and a transformer-based model for mask instance prediction. The 3D CNN backbone takes the voxelized input point cloud scene as input, and outputs multi-level feature maps, while the transformer decoder takes the multi-level feature maps to refine a set of queries through self and cross-attention. The final refined queries are used to predict instance masks. The 3D proposal network predicts a set $K_{3D} \in \mathbb{N}$ of 3D mask proposals $M \in \mathbb{Z}_2^{K_{3D} \times N}$ for a given point cloud $P \in \mathbb{R}^{4 \times N}$ with $N$ points in homogeneous coordinate system, where $\mathbb{Z}_2 = \{0, 1\}$.

### 4.3 LOW GRANULARITY (LG) LABEL-MAPS

As discussed earlier, the focus of our approach is to generate fast and accurate open-vocabulary labels for the generated 3D proposals. Instead of relying on computationally intensive 2D segmentation, we propose a 2D detection-based approach in our pipeline. For every RGB image $\mathcal{I}_i$ we generate a set of $K_{b,i}$ bounding boxes $\mathbf{B}_i = \{(b_j, c_j) \mid b_j \in \mathbb{R}^4, \quad c_j \in \mathbb{N}, \quad \forall j \in (1, ..., K_{b,i})\}$ using an open-vocabulary 2D object detector, where $b_j$ are the bounding boxes coordinates while $c_j$ is its predicted label. We assign a weight $w_j = b_j^H + b_j^W$ for each output bounding box $b_j$, where $b_j^H$ and $b_j^W$ are the bounding box's height and width, respectively. The weights represent the bounding box size and help determine the order of bounding boxes when used to construct the LG label maps.

After obtaining the 2D object detections, we represent the output of each 2D image frame $\mathcal{I}_i$ as an LG label map $\mathcal{L}_i \in \mathbb{Z}^{W \times H}$. To construct $\mathcal{L}_i$, we start by initializing all of its elements as **-1**. In our notation, **-1** represents no class and is ignored during prediction. Next, we sort all bounding

boxes following their weights and replace the region of the bounding box $b_j$ with its corresponding predicted class label $c_j$ in the label map, starting with the bounding box with the highest weight. The weight $w_j$ choice is motivated by the fact that if two objects of different sizes appear in the same direction the camera is pointing to, the small object is visible in the image if it is closer to the camera than the large object.

## 4.4 Accelerated Visibility Computation (VACC)

In order to associate 2D label maps with 3D proposals, we compute the visibility of each 3D mask. To this end, we propose a fast approach that is able to compute 3D mask visibility within frames via tensor operations which are highly parallelizable.

Given $K_f \in \mathbb{N}$ 2D RGB frames $\mathcal{I} = \{\mathcal{I}_i \in \mathbb{R}^{3 \times W \times H} \mid \forall i \in (1, ..., K_f)\}$ associated with the 3D scene $P$, along with their intrinsic matrices $I \in \mathbb{R}^{N_f \times 4 \times 4}$ and extrinsic matrices $E \in \mathbb{R}^{N_f \times 4 \times 4}$. We denote the projection of the 3D point cloud $P$ on a frame with index $i$ $P_i^{2D} \in \mathbb{R}^{4 \times N}$, and it can be computed as follows $P_i^{2D} = I_i \cdot E_i \cdot P$. After stacking the projection operations of all frames, the projection of the scene onto all frames can be computed with a GPU in a single shot as follows

$$P^{2D} = (I \star E) \cdot P$$

where $\star$ is batch-matrix multiplication, $\cdot$ is matrix multiplication, and $P^{2D} \in \mathbb{R}^{N_f \times 4 \times N}$ defined as $P^{2D} = [P_1^{2D}, ..., P_{N_f}^{2D}]$.

Furthermore, we compute the visibility $V^f \in \mathbb{Z}_2^{N_f \times N}$ of all projected points within all frames as follows

$$V^f = \mathbb{1}(0 < P_x^{2D} < W) \odot \mathbb{1}(0 < P_y^{2D} < H)$$

where $\mathbb{1}$ is the indicator function, $W$ and $H$ are the image width and height respectively , $\odot$ is element wise multiplication, $P_x^{2D} \in \mathbb{R}^{N_f \times N}$ and $P_y^{2D} \in \mathbb{R}^{N_f \times N}$ are the projected 3D points $x$ and $y$ coordinates on all $N_f$ frames, respectively.

For occlusion, we define another visibility matrix as $V^d \in \mathbb{Z}_2^{N_f \times N}$ that is computed as follows

$$V^d = \mathbb{1}(|P_z^{2D} - D_z| < \tau_{depth})$$

where $D_z \in \mathbb{R}^{N_f \times N}$ is the real depth of the 3D point cloud obtained from the depth maps, while $P_z^{2D} = P_{1 \leq i \leq N_f, j=3, 1 \leq k \leq N}^{2D}$ is the depth obtained from projecting the point cloud $P$, and $|\cdot|$ is the absolute value. Finally, the per 3D mask proposal visibility $\{V \in \mathbb{R}^{N_f \times K_{3D}}, \quad 0 \leq V \leq 1\}$, can be computed in a single shot using a GPU as follows

$$V = \left((V^f \odot V^d) \cdot M^T\right) \odot M_{count}^{-1}$$

where $M \in \mathbb{Z}_2^{K_{3D} \times N}$ is the matrix of 3D mask proposals generated by the 3D proposal network, and $M_{count} \in \mathbb{N}^{K_{3D}}$ is the count of points per mask,

The percentage of visibility of a mask $j$ in a frame $i$ is represented with the element $V_{i,j} \in V$.

## 4.5 Multi-View Prompt Distribution (MVPDist)

Given an RGB sequence $\mathcal{I}$ and their corresponding LG label-maps $\mathcal{L} = \{\mathcal{L}_i \mid \mathcal{L}_i \in \mathbb{Z}^{W \times H}, \quad \forall i \in (1, ..., K_f)\}$. We define the label distribution $\mathcal{D}_j \in \mathbb{Z}^{N_{dist}^j}$ of a 3D mask proposal $M_j$ as

$$\mathcal{D}_j = \{\mathcal{L}_i \left[P_{i,x}^{2D} \cdot M_{ji}, P_{i,y}^{2D} \cdot M_{ji}\right] \mid \forall i \in \mathcal{P}_k\}$$

where $M_{ji} = V_i^d \cdot V_i^f \cdot M_j$ is the mask for non-occluded in-frame points that belong to the $j^{th}$ instance, $\mathcal{P}_k$ is the set of frame indices where the $j^{th}$ 3D mask has top-k visibility, computed using the visibility matrix $V$, $P_{i,x}^{2D}$ and $P_{i,y}^{2D}$ are the projected x and y coordinates of the point cloud scene onto the $i_{th}$ frame, respectively, while $[\cdot, \cdot] : \mathbb{Z}^{W \times H} \mapsto \mathbb{Z}^n$ is a coordinate-based selection operator with $n$ as an arbitrary natural number.

We define the probability for the mask $M_j$ to be assigned to a class $c$ as the occurrence of class $c$ within the distribution $\mathcal{D}_j$; we demonstrate the approach in Figure 3 for one 3D proposal. We assign the class with the highest probability to the mask $M_j$.

Figure 3: **Multi-View Prompt Distribution (MVPDist).** After creating the LG label maps for all frames, we select the top-k label maps based on the 2D projection of the 3D proposal. Using the (x, y) coordinates of the 2D projection, we choose the labels from the LG label maps to generate the MVPDist. This distribution predicts the ID of the text prompt with the highest probability.

## 4.6 INSTANCE PREDICTION CONFIDENCE SCORE

We propose in this section a way to score class-agnostic 3D instance mask proposals by leveraging 2D information from bounding boxes along with 3D information from 3D masks. Our rationale is that a good mask should lead to a high class and mask confidence, unlike previous Open-Vocabulary 3D instance segmentation methods, which use only class confidence. To achieve this, we define the score of a 3D mask proposal as $s_m = s_{IoU} \cdot s_{class}$, where $s_{class}$ is the probability of the class with the highest occurrence in MVPDist, while $s_{IoU}$ is the average IoU across multi-views between the bounding box of the projected 3D mask and the bounding box with the highest IoU generated with a 2D object detector in the view (More details are in the appendix).

## 5 EXPERIMENTS

**Datasets:** We conduct our experiments using the ScanNet200 Rozenberszki et al. (2022) and Replica Straub et al. (2019) datasets. Our analysis on ScanNet200 is based on its validation set, comprising 312 scenes. For the 3D instance segmentation task, we utilize the 200 predefined categories from the ScanNet200 annotations. ScanNet200 labels are categorized into three subsets—head (66 categories), common (68 categories), and tail (66 categories)—based on the frequency of labeled points in the training set. This categorization allows us to evaluate our method's performance across the long-tail distribution, underscoring ScanNet200 as a suitable evaluation dataset. Additionally, to assess the generalizability of our approach, we conduct experiments on the Replica dataset, which has 48 categories. For the metrics, we follow the evaluation methodology in ScanNet Dai et al. (2017) and report the average precision (AP) at two mask overlap thresholds: 50% and 25%, as well as the average across the overlap range of [0.5:0.95:0.05].

**Implementation details:** We use RGB-depth pairs from the ScanNet200 and Replica datasets, processing every 10th frame for ScanNet200 and all frames for Replica, maintaining the same settings as OpenMask3D for fair comparison. To create LG label maps, we use the YOLO-World Cheng et al. (2024) extra-large model for its real-time capability and high zero-shot performance. We use Mask3D Schult et al. (2023) with non-maximum suppression to filter proposals similar to Open3DIS Nguyen et al. (2024), and avoid DBSCAN Ester et al. (1996) to prevent inference slowdowns. We use a single NVIDIA A100 40GB GPU for all experiments.

## 5.1 RESULTS ANALYSIS

**Open-Vocabulary 3D instance segmentation on ScanNet200:** We compare our method's performance against other approaches on the ScanNet200 dataset in Table 1. We indicate whether each method uses 2D instances to generate 3D proposals and whether SAM is used for labeling the 3D masks. Our method achieves state-of-the-art performance with proposals from only a 3D instance segmentation network compared to methods from two settings *(i)* 3D mask proposals from only a 3D instance segmentation network *(ii)* using a combination of 3D mask proposals from a 3D network and 2D instances from a 2D segmentation model. Additionally, our method is ∼ 16× faster compared to state-of-the-art Open3DIS.

Table 1: **State-of-the-art comparison on ScanNet200 validation set.** We use Mask3D trained on the ScanNet200 training set to generate class-agnostic mask proposals. Our method demonstrates better performance compared to those that generate 3D proposals by fusing 2D masks and proposals from a 3D network (highlighted in gray in the table), while it outperforms state-of-the-art methods by a wide margin under the same conditions using only proposals from a 3D network.

| Method | 3D proposals from 2D masks | SAM for 3D mask labeling | mAP | mAP50 | mAP25 | head | comm | tail | time/scene (s) |
|---|---|---|---|---|---|---|---|---|---|
| Mask3D (Closed Vocab.) | × | × | 26.9 | 36.2 | 41.4 | 39.8 | 21.7 | 17.9 | 13.41 |
| SAM3D | ✓ | × | 6.1 | 14.2 | 21.3 | 7.0 | 6.2 | 4.6 | 482.60 |
| OVIR-3D | ✓ | × | 13.0 | 24.9 | 32.3 | 14.4 | 12.7 | 11.7 | 466.80 |
| Open3DIS | ✓ | × | 23.7 | 29.4 | 32.8 | 27.8 | 21.2 | 21.8 | 360.12 |
| OpenScene (2D Fusion) | × | × | 11.7 | 15.2 | 17.8 | 13.4 | 11.6 | 9.9 | 46.45 |
| OpenScene (3D Distill) | × | × | 4.8 | 6.2 | 7.2 | 10.6 | 2.6 | 0.7 | **0.26** |
| OpenScene (2D-3D Ens.) | × | × | 5.3 | 6.7 | 8.1 | 11.0 | 3.2 | 1.1 | 46.78 |
| OpenMask3D | × | ✓ | 15.4 | 19.9 | 23.1 | 17.1 | 14.1 | 14.9 | 553.87 |
| Open3DIS | × | × | 18.6 | 23.1 | 27.3 | 24.7 | 16.9 | 13.3 | 57.68 |
| **Open-YOLO 3D (Ours)** | × | × | **24.7** | **31.7** | **36.2** | 27.8 | **24.3** | 21.6 | 21.8 |

Table 2: **State-of-the-art comparison on Replica dataset.** We use Mask3D trained on the Scan-Net200 training set to generate class-agnostic mask proposals. We show that our method generalizes better than state-of-the-art methods under the same setting with proposals from 3D networks only.

| Method | 3D proposals from 2D masks | SAM for 3D mask labeling | mAP | mAP50 | mAP25 | time/scene (s) |
|---|---|---|---|---|---|---|
| OVIR-3D | ✓ | × | 11.1 | 20.5 | 27.5 | 52.74 |
| Open3DIS | ✓ | × | 18.5 | 24.5 | 28.2 | 187.97 |
| OpenScene (2D fusion) | × | × | 10.9 | 15.6 | 17.3 | 317.327 |
| OpenScene (3D Distill) | × | × | 8.2 | 10.5 | 12.6 | **4.29** |
| OpenScene (2D-3D Ens.) | × | × | 8.2 | 10.4 | 13.3 | 320.127 |
| OpenMask3D | × | ✓ | 13.1 | 18.4 | 24.2 | 547.32 |
| Open3DIS | × | × | 14.9 | 18.8 | 23.6 | 35.08 |
| **Open-YOLO 3D (Ours)** | × | × | **23.7** | **28.6** | **34.8** | 16.6 |

**Generalizability to unseen dataset:** To test the generalizability of our method, we use the 3D proposal network pre-trained on the ScanNet200 training set and evaluate on the Replica dataset; the results are shown in Table 2. Our method shows competitive performance with $\sim 11 \times$ speedup against state-of-the-art models Nguyen et al. (2024), which use CLIP features. We highlight that OpenScene (3D Distill) achieves the highest speed by using a trained 3D U-Net to directly predict per-point CLIP features in under a second, whereas 2D prior-based methods are slower but offer better generalization.

**Performance with given 3D masks:** We further test our method for 3D proposal prompting from text against existing methods in the literature in the case of ground truth 3D masks and report the results in Table 5. For Mask3D, oracle masks are assigned class predictions from matched predicted masks using Hungarian matching. For open-vocabulary methods, we use ground-truth masks as input proposals. We show the results in Table 5 that MVPDist can out-

Table 5: **Comparative results with oracle masks on ScanNet200 dataset.** Our MVPDist significantly outperforms state-of-the-art methods using CLIP features. We show that MVPDist with YOLO-WORLD predictions can use multi-view information to achieve high zero-shot capabilities.

| Method | 2D OV Prior | mAP | head | comm | tail |
|---|---|---|---|---|---|
| *Closed Vocab.* Mask3D | None | 35.5 | 55.2 | 27.2 | 22.2 |
| OpenScene | 2D LSeg | 11.8 | 26.9 | 5.2 | 1.7 |
| OpenScene | 2D OpenSeg | 22.9 | 26.2 | 22.0 | 20.2 |
| OpenMask3D | Mask wise CLIP | 29.1 | 31.1 | 24.0 | 32.9 |
| Open3DIS | Point wise CLIP | 30.9 | 33.1 | 25.1 | 35.4 |
| **Open-YOLO 3D (Ours)** | MVPDist | **39.6** | **43.3** | **36.8** | **38.5** |

perform CLIP-based approaches in retrieving the correct 3D proposal masks from text prompts, due to the high zero-shot performance achieved by state-of-the-art open-vocabulary 2D object detectors.

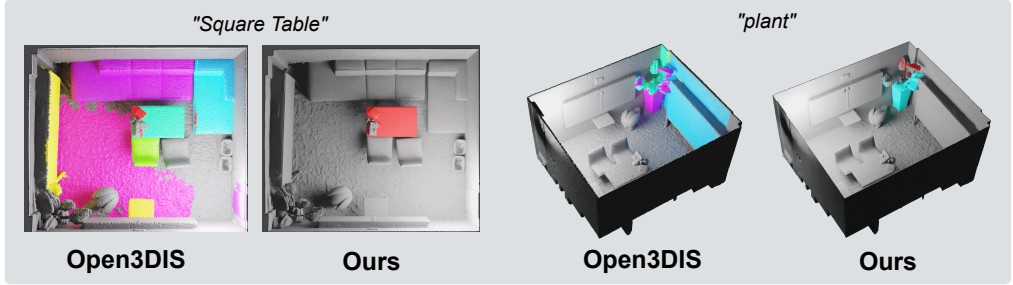

Figure 4: **Qualitative results on scene *office0* in the Replica dataset.** We show instances with a confidence score above 0.5 for both methods. We show that our method is much more precise when segmenting the object in the text compared to state-of-the-art method Open3DIS.

Table 3: **Effect of crop generation prior on OpenMask3D.** R1 corresponds to the original Open-Mask3D, while we show the effect of replacing SAM with a 2D object detector in rows R2 to R4.

| Row ID ↓ | 2D Prior | Labeling method | MVPDist | VAcc | mAP | time/scene (s) |
|---|---|---|---|---|---|---|
| R1 | SAM+CLIP | Clip features | × | × | 33.0 | 675.6 |
| R2 | YoloV8+CLIP | Clip features | × | × | 21.1 | 440.40 |
| R3 | RT-DETR+CLIP | Clip features | × | × | 28.4 | 487.95 |
| R4 | YoloWorld+CLIP | Clip features | × | × | 32.5 | 384.29 |

**Generalizability to out of distribution classes:** We conduct an experiment similar to Open-Mask3D, and report the results in Table 6. In this experiment, we evaluate our model's performance using Mask3D pretrained on the ScanNet20 dataset to generate class-agnostic masks. The results are reported for Novel classes, comprising 147 classes from ScanNet200 that differ from the 18 classes in ScanNet20, and for Base classes, which consist of 47 classes that overlap with those in ScanNet20. Our method demonstrates better performance on both Novel and Base classes.

Table 6: **Comparative results on ScanNet200 validation set with Mask3D pretrained on ScanNet20 training set.** We show the results with our method on Novel and Base classes as defined in OpenMask3D paper Takmaz et al. (2023), with class agnostic masks generated by Mask3D pre-trained on ScanNet20.

| Model | mAP | $mAP_{base}$ | $mAP_{novel}$ | time/scene (s) |
|---|---|---|---|---|
| OpenMask3D | 12.6 | 14.3 | 11.9 | 511.41 |
| Open3DIS | 19.0 | 25.8 | 16.5 | 312.11 |
| OpenYolo3D | **19.6** | **26.1** | **17.1** | **22.7** |

**Ablation over Replica dataset:** To evaluate object detectors as alternatives to SAM for generating crops in 3D CLIP feature aggregation, we tested three detectors on the Replica dataset, using ground truth 3D mask proposals to compare labeling performance.

Table 7: **Ablation study on deducted components and their impact on mAP and inference time.**

| Row ID | Deducted Components | CLassification Prior | mAP | Time (s) |
|---|---|---|---|---|
| 0 | Ours - (MVPDist & Vacc) | YoloWorld | 19.9 | 392.02 |
| 1 | Ours - (MVPDist & Vacc) | CLIP features | 32.5 | 396.89 |
| 2 | Ours - Vacc | YoloWorld | 46.2 | 376.42 |
| 3 | Ours | YoloWorld | **46.2** | **17.86** |

Results are shown in Table 3, rows **R2** to **R4**, with **R1** presenting baseline results from OpenMask3D Takmaz et al. (2023). We generate class-agnostic bounding boxes with an object detector, then assign the highest IoU bounding box to each 3D instance as a crop, selecting the most visible views. For 3D CLIP feature aggregation, we follow OpenMask3D's approach, aggregating features from multiple levels and views. These experiments show that YOLO-World can generate crops nearly as good as SAM but with significant speed improvements. To show the effect of adding our contributions on the overall performance, we report the results in table 7. We show in Row **2** the zero-shot performance of our proposed Multi-View Prompt Distribution with LG label maps compared to (*i*) Row **0** where we assign the 3D mask a class label based on the YoloWorld prediction with the highest IoU overlap and confidence across the views. This experiment in row 0 relies solely on YoloWorld predictions (*ii*) using the class prediction obtained from the CLIP feature of the corresponding matched bounding box. Additionally, Row **3** shows speed improvements using the GPU for 3D mask visibility computation.

Table 4: **Effect of number of top-k views on the labeling performance with High Granularity (HG) and Low-Granularity (LG) label maps**. HG Label maps are generated by prompting SAM with the bounding boxes used to generate LG label maps.

| Topk ($\rightarrow$) Map type ($\downarrow$) | 1 mAP/ time(s) | 10 mAP/ time(s) | 20 mAP/ time(s) | 30 mAP/ time(s) | 40 mAP/ time(s) |
|---|---|---|---|---|---|
| HG label maps | 16.3/113.93 | 22.5 / 115.16 | 23.9 / 114.76 | 24.4 / 115.89 | 24.5 / 115.92 |
| LG label maps | 16.3 / 20.7 | 22.8 / 21.6 | 24.3 / 21.7 | 24.6 / 22.1 | 24.7 / 22.2 |

Table 8: **Results with 3D masks lifted using hierarchical agglomerative clustering proposed in Open3DIS.** For 2D mask generation, we use G-DINO and SAM; then we compare the class label prediction capabilities for the 3D aggregated masks between point-wise features with clip as in Open3DIS and our proposed Multi-View prompt distribution with YoloWorld.

| Model | Proposal method | mAP | mAP50 | mAP25 | mAP$_{tail}$ | mAP$_{com}$ | mAP$_{head}$ | Time/scene (s) |
|---|---|---|---|---|---|---|---|---|
| Open3DIS | 3D-Net + 2D-Net | 23.7 | 29.4 | 32.8 | 27.8 | 21.2 | 21.8 | 360.12 |
| Ours | 3D-Net + 2D-Net | **25.6** | **31.9** | **36.1** | **27.9** | **24.2** | **24.7** | **303.41** |

**Top K analysis:** We show in Table 4 that naively using YOLO-World with only one label-map with the highest visibility per 3D mask proposal results in sub-optimal results and using top-K label-maps can result in better predictions as the distribution can provide better estimate across multiple frames, since YOLO-World is also expected to make misclassifications in some views while generating correct ones in others. This approach assumes that YOLO-World makes a correct class prediction for the same 3D object in multiple views for it to be effective.

**High-Granularity (HG) vs. Low-Granularity (LG):** Table 4 shows that using SAM to generate HG label maps slightly reduces mAP, and slows down the inference by $\sim$ 5 times. This is due to the nature of projected 3D instances into 2D, where the projection already holds 2D instance information as shown in Figure 3, and SAM would just result in redundancy in the prediction.

**How does OpenYolo3D compare to other clip-based methods under different scenarios?** In scenarios such as robot navigation where the RGB-D sequence and point cloud change over time or when the user has a set of classes that they want to extract in a single shot, we show through our experiments that OpenYolo3D significantly outperforms other methods, such as OpenMask3D Takmaz et al. (2023) and Open3DIS Nguyen et al. (2024), where the entire pipeline must be executed from scratch for a set of text prompt. However, in static scenes where users want to prompt objects sequentially, clip-based models can run once to extract clip features from multiple views, allowing them to respond to new prompts within milliseconds. In contrast, our method, with stored visibility matrix and visual features from the 2D detector backbone, takes 1 to 2 seconds for each new prompt under the later scenario for small scenes.

**Limitations:** Our method makes use of a 3D proposal network only for proposal generation in order to reach high speed. Other proposal generation methods Lu et al. (2023); Nguyen et al. (2024) fuse 2D instance masks from a 2D instance segmentation methods to generate rich 3D proposals even for very small objects, which are generally overlooked by 3D proposal networks like Mask3D Schult et al. (2023) due to low resolution in 3D. Thus, fast 2D instance segmentation models like FastSAM Zhao et al. (2023) can be used to generate 3D proposals from the 2D images, which might further improve the performance of our method.

## 6 CONCLUSION

We present Open-YOLO 3D, a novel and efficient open-vocabulary 3D instance segmentation method, which makes use of open-vocabulary 2D object detectors instead of heavy segmentation models. Our approach leverages a 2D object detector for class-labeled bounding boxes and a 3D instance segmentation network for class-agnostic masks. We propose to use MVPDist generated from multi-view low granularity label maps to match text prompts to 3D class agnostic masks. Our proposed method outperforms existing techniques, with gains in mAP and inference speed. These results show a new direction toward more efficient open-vocabulary 3D instance segmentation models.

**Acknowledgement** The computations were enabled by resources provided by NAISS at Alvis partially funded by Swedish Research Council through grant agreement no. 2022-06725, LUMI hosted by CSC (Finland) and LUMI consortium, and by Berzelius resource provided by the Knut and Alice Wallenberg Foundation at the NSC.

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
