# OPEN-YOLO 3D: TOWARDS FAST AND ACCURATE OPEN-VOCABULARY 3D INSTANCE SEGMENTATION -SUPPLEMENTARY MATERIAL-

**Mohamed El Amine Boudjoghra**
TUM, MBZUAI

**Angela Dai**
TUM

**Jean Lahoud**
MBZUAI

**Hisham Cholakkal**
MBZUAI

**Rao Muhammad Anwer**
MBZUAI, Aalto University

**Salman Khan**
MBZUAI, ANU

**Fahad Shahbaz Khan**
MBZUAI, Linköping University

## 1 COMPLEXITY ANALYSIS OF VACC

The in-frame 3D mask visibility in OpenMask3D is calculated by iterating over each mask and frame to count the visible points within each frame. This process involves $N$ 3D masks, $M$ frames, and $P$ points, leading to a computational and time complexity of $O(N \times M \times P)$. However, the use of sequential loops restricts the potential for efficient parallelization.

In our proposed method, **VAcc**, we reformulate the visibility computation using tensor operations, enabling parallelization across multiple cores. While maintaining the same computational complexity $O(N \times M \times P)$, it reduces time complexity to $O(N \times M \times P/c)$, where $c$ is the number of cores. This makes VAcc a faster alternative to the method used in OpenMask3D.

## 2 EFFECT OF OCCLUSION AND LIGHT CONDITIONS

### 2.1 OCCLUSION ANALYSIS

This analysis evaluates model robustness to occlusion using the testing protocol from Naseer et al. (2021). Images are systematically occluded by dropping patches (1% of image size) centered on pixels from 2D projections of 3D ground truth instance masks. The experiments include:

**Salient Patch Drop**: Simulates the occlusion of semantically important foreground objects by removing patches from 2D masks generated from instance masks, while excluding areas corresponding to walls, floors, and ceilings. To ensure uniform evaluation, larger masks have a proportionally greater number of patches removed, effectively testing the impact of occlusion on detector performance.

**Non-Salient Patch Drop**: Focuses on background occlusion by removing patches from regions corresponding to walls, floors, and ceilings, thereby testing the detector's robustness to non-critical occlusions.

These experiments assess how meaningful and arbitrary occlusions affect object detection under extreme conditions.

Table 1: **Occlusion analysis results for salient and non-salient occlusions.**

| Percentage (%) | 0 | 5 | 30 | 55 | 80 | 95 | 100 |
|---|---|---|---|---|---|---|---|
| **Salient** | 46.2 | 40.6 | 18.1 | 8.0 | 7.0 | 0.5 | 0.0 |
| **Non-Salient** | 46.2 | 44.5 | 39.4 | 39.7 | 38.0 | 36.1 | 35.3 |

We show that our method is robust to moderate levels of salient occlusion where the important objects are occluded while being extremely robust when the non-salient objects are occluded.

## 2.2 ANALYSIS OF EXTREME LIGHTING CONDITIONS

We test the model's robustness to different levels of lighting conditions, starting with 0.05 intensity (almost black) up to 2.5 intensity (extremely bright). A brightness level of 1 corresponds to the original lighting conditions. The robustness results demonstrate that the pipeline performs well under moderate fluctuations in light intensity but encounters challenges in extremely low-light conditions.

Table 2: **Performance under different lighting conditions.**

| Brightness | 0.05 | 0.25 | 0.5 | 0.75 | 1 | 1.25 | 1.5 | 1.75 | 2.5 |
|---|---|---|---|---|---|---|---|---|---|
| **mAP (ours)** | 15.6 | 39.3 | 40.8 | 44.7 | 46.2 | 43.0 | 41.1 | 40.0 | 35.6 |

Table 3: **Results on Base/Novel replica splits with Mask3D as the proposal network.**

| Method | mAP | mAP$_{base}$ | mAP$_{novel}$ |
|---|---|---|---|
| Open3DIS | 18.2 | 19.9 | 10.8 |
| **Ours** | **23.7** | **26.4** | **11.9** |

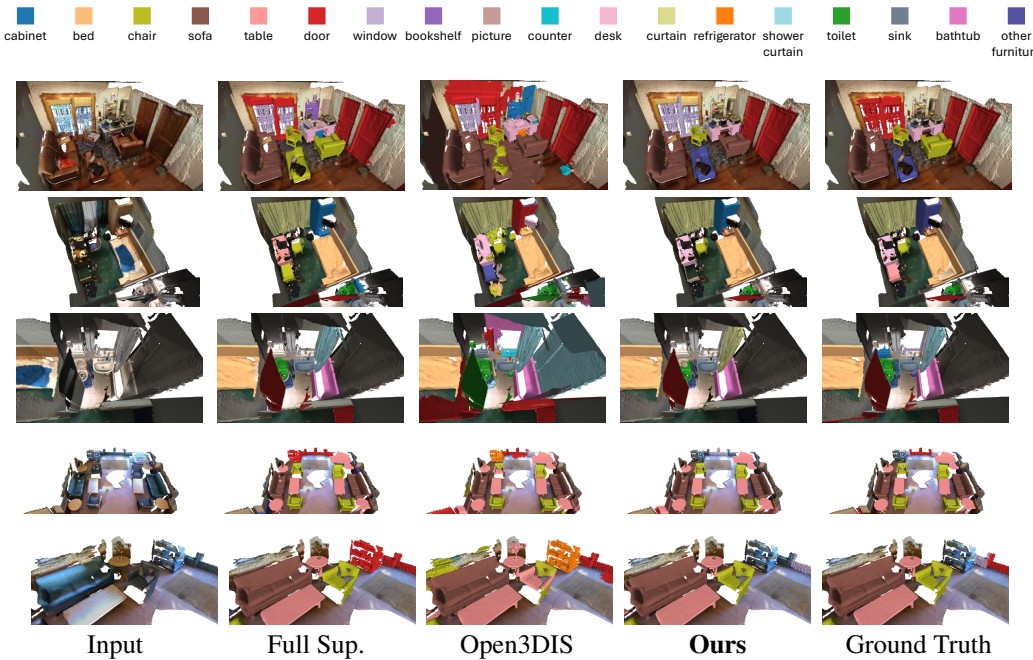

Figure 1: **Comparison of our method, OpenYOLO3D (second column from the right), against other methods using ScanNet20 classes as text prompts (excluding "floor" and "wall"). We use Mask3D trained on ScanNet200 labels for class-agnostic mask generation for our method.** We compare against Mask3D fully supervised on ScanNet20 labels and the state-of-the-art 3D instance segmentation model, Open3DIS. The results demonstrate that our method performs significantly better than Open3DIS (with Mask3D as 3D network +G-DINO as 2D Network) in 3D mask classification. Additionally, our method performs comparably to the fully supervised Mask3D classifier on the 18 classes.

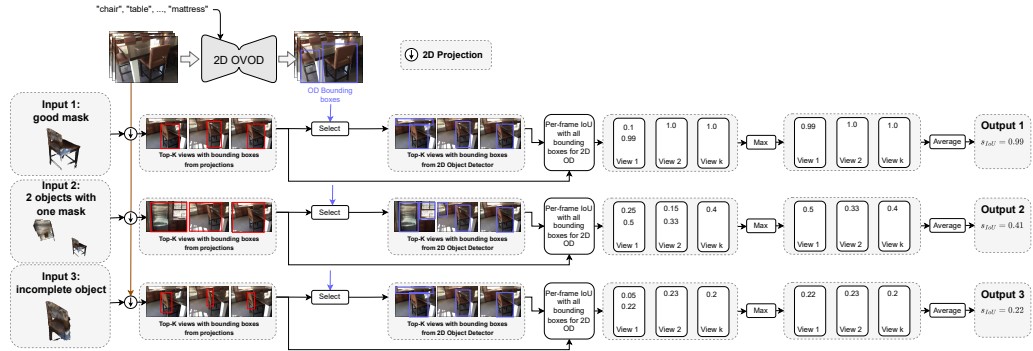

Figure 2: **Additional details on how IoU score $s_{IoU}$ is computed.** We show that our method can provide a reliable mask score using Intersection Over Union (IoU) between the bounding boxes estimated using the 3D cropped instance 2D projection and the bounding boxes from a 2D object detector. We also demonstrate that it covers all three cases of different 3D mask proposals.

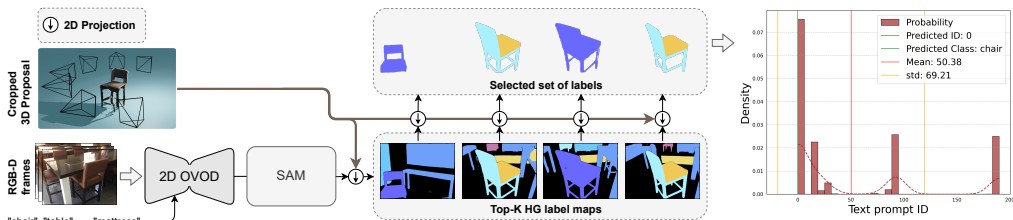

Figure 3: **Additional details on how the High-Granularity label maps (HG label maps) are constructed.** To get a detailed per-pixel label for the label maps, we prompt SAM with bounding boxes to get pixel-wise masks, and then we follow the same approach as in the paper to construct the label maps. The multi-View Prompt Distribution is estimated as described in the main paper

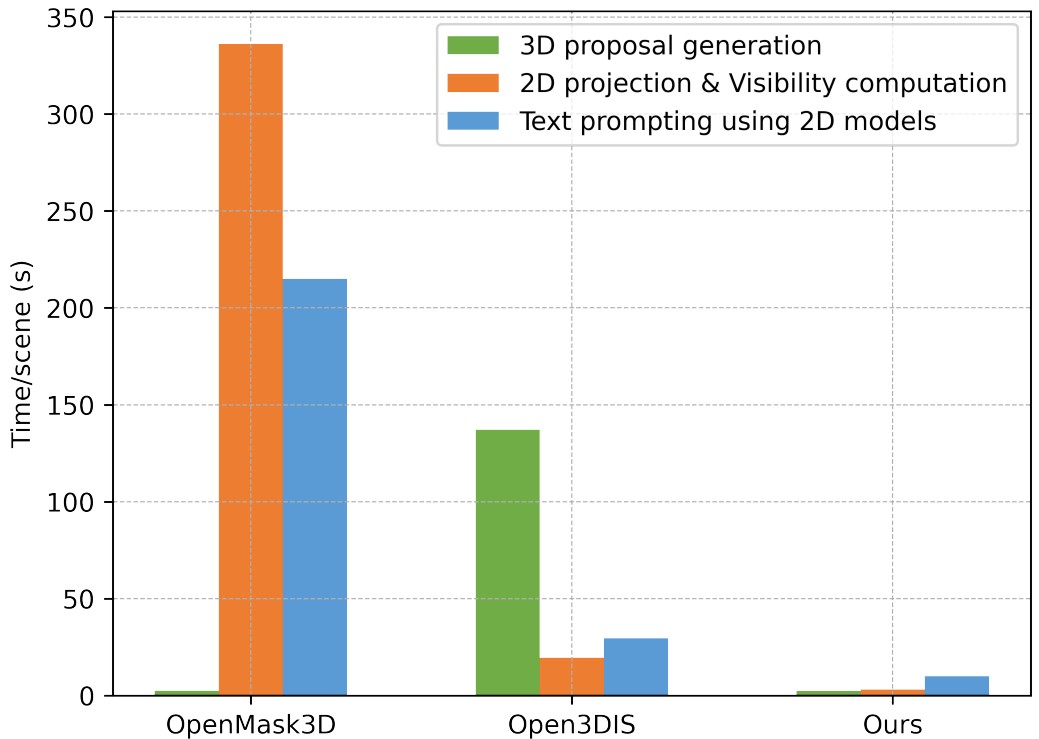

Figure 4: **Time breakdown of the average time for Open-Vocabulary 3D instance segmentation over the replica dataset.** We report the time for each step that makes the whole Open-Vocabulary 3D instance segmentation pipeline for our method and the two most recent methods, OpenMask3D and Open3DIS. **3D proposal generation** is the step of generating class agnostic mask proposals for the objects in the 3D point cloud scene **2D projection & Visibility computation** is the step of projection of the point cloud scene onto the 2D frames, and computing the visibility of each mask proposal in the 2D frames. **Text prompting with 2D models** is the step of inference from the 2D models to prompt the 3D class agnostic masks with text.

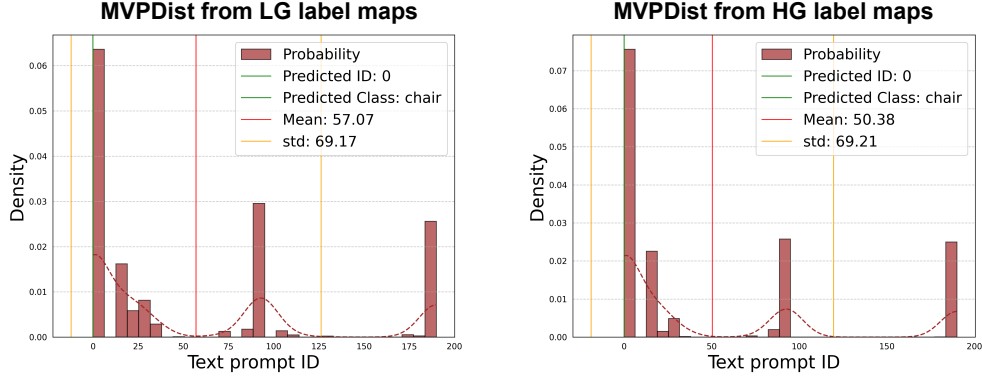

Figure 5: **Comparison between MVPDist with HG label maps and with LG label maps.** We show that using HG label maps instead of LG label maps slightly increases the prediction class score but results in the same prediction. Thus, LG can give a similar performance with around 5 times speed up in terms of inference time.

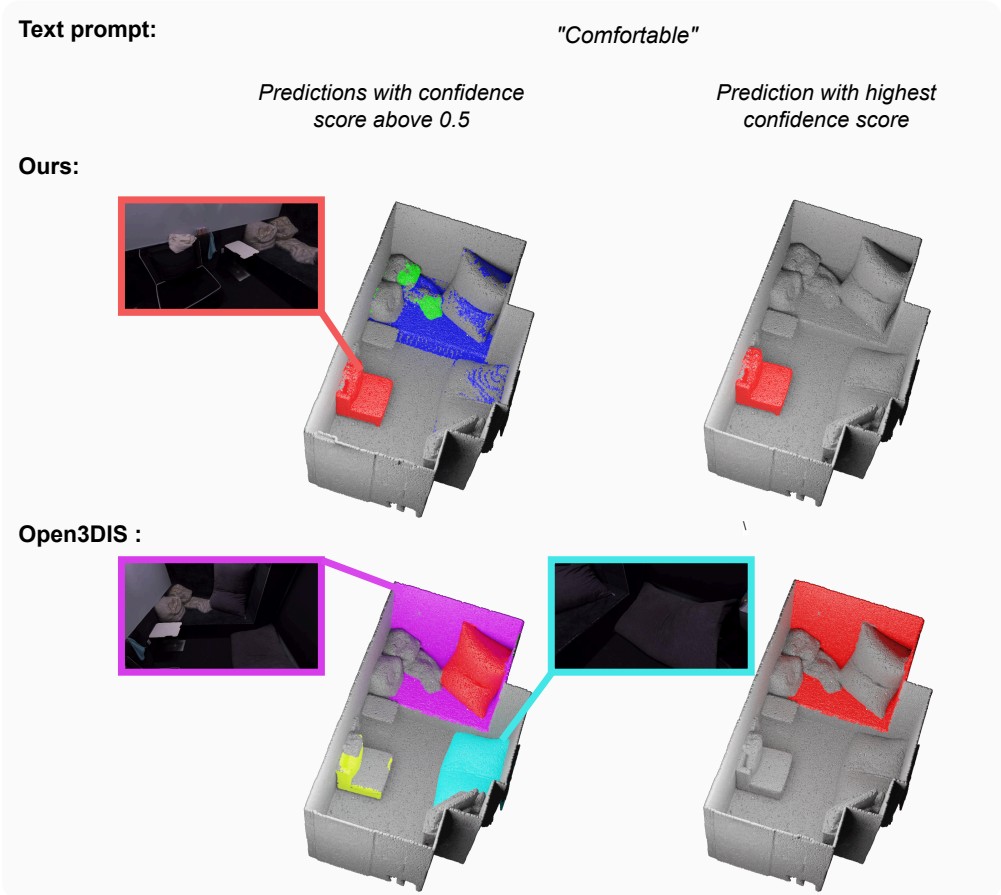

Figure 6: **Comparison between our method and Open3DIS.** We show that our method is more precise than Open3DIS.

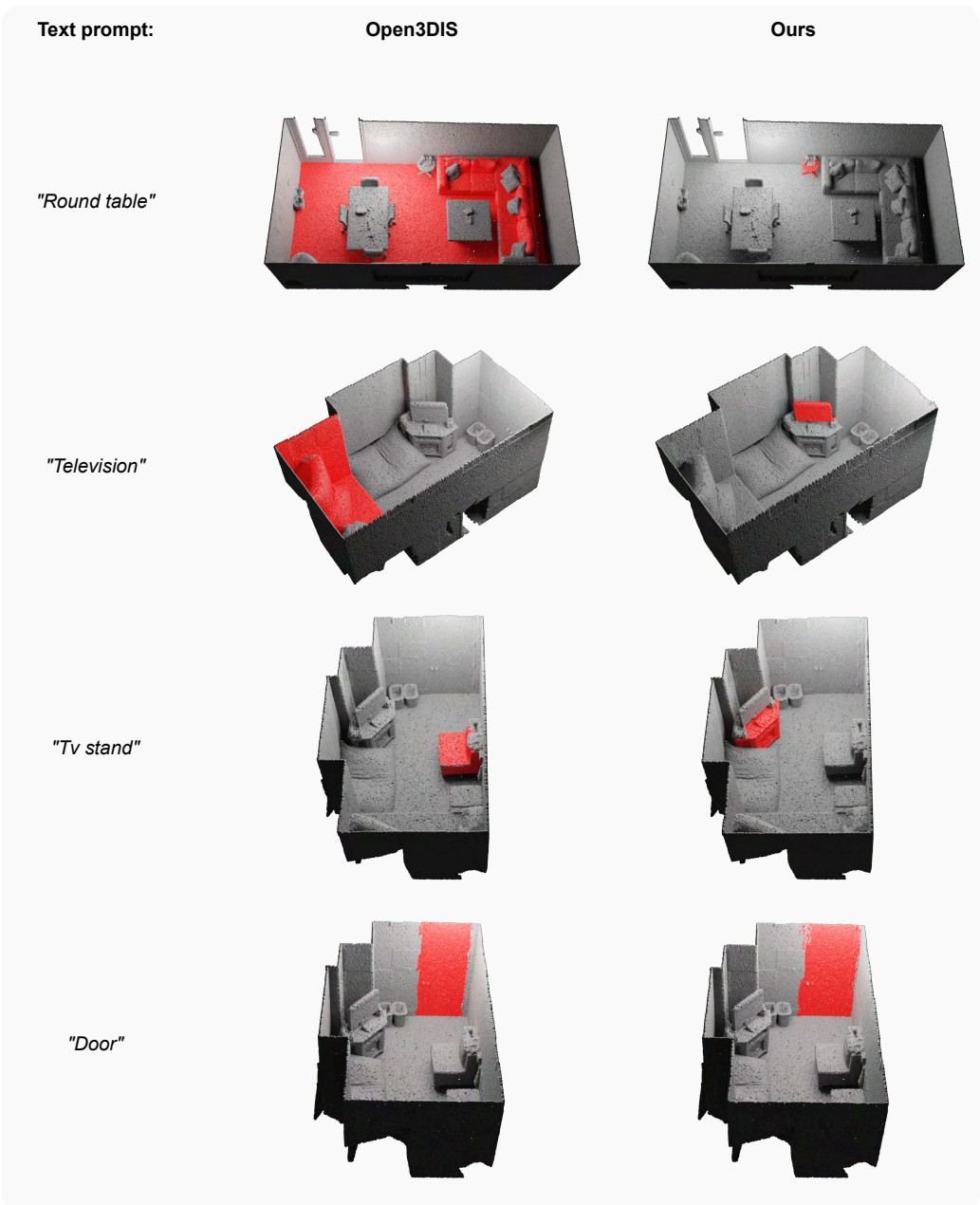

Figure 7: **Comparison of prediction with the highest score between our method and Open3DIS on replica dataset.**

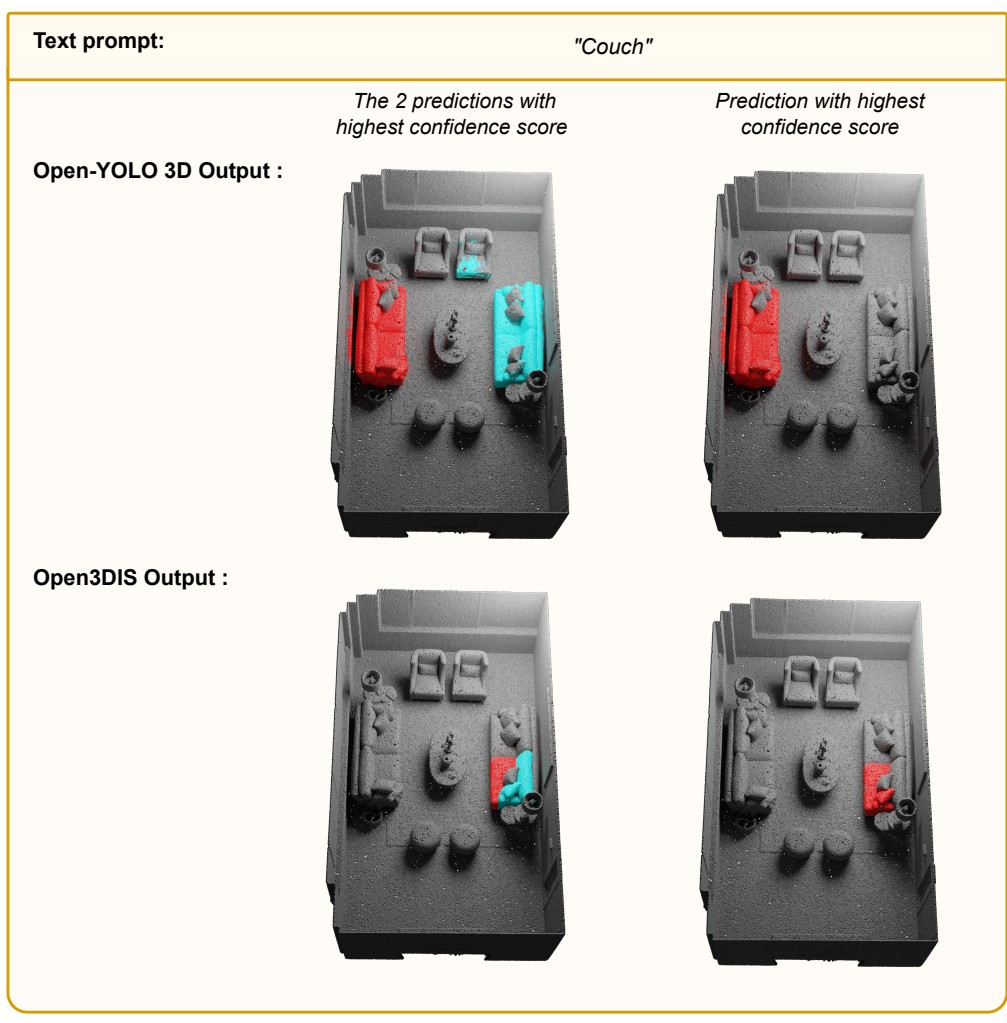

Figure 8: **Scoring quality comparison between ours and Open3DIS. (Top)** We show on the left the two masks with top confidence scores $s_m$ estimated using our approach, notice that the second mask in blue has a portion that extends to another instance, which makes it worse in than the red mask. Our scoring technique captures this detail while scoring the mask. **(Bottom)** We show on the left the two masks with the highest scores by Open3DIS. Notice that using only semantic scores results in wrong mask confidence since it is aggregated from 2D clip features, where points with the most visibility across views contribute more than others to the final aggregated 3D clip features. Thus, this type of scoring assigns very high scores to bad masks that have points with more visibility.

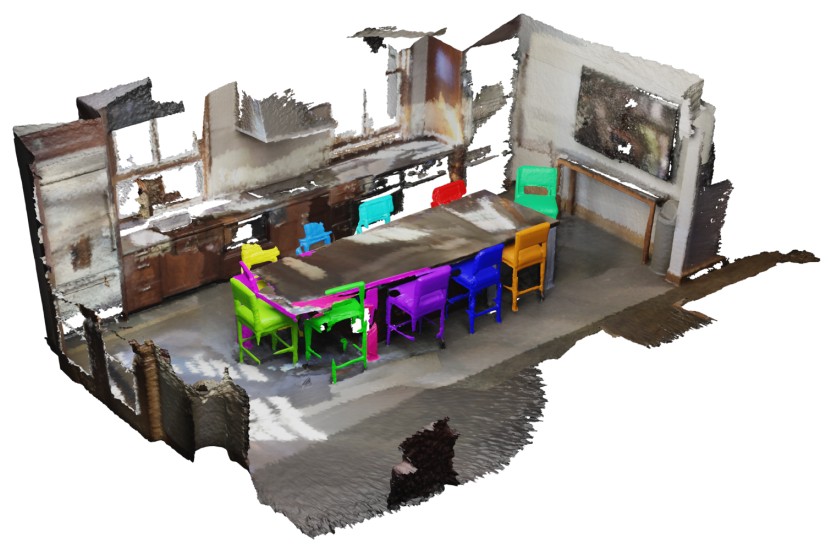

Figure 9: **Additional qualitative results on ScanNet200.** with text prompt *"Chair"*.

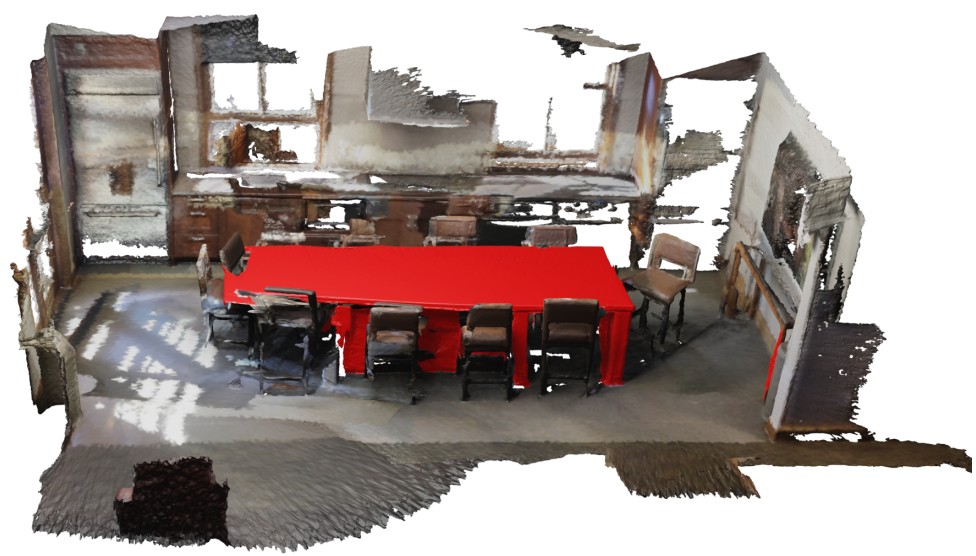

Figure 10: **Additional qualitative results on ScanNet200.** with text prompt *"Table"*.

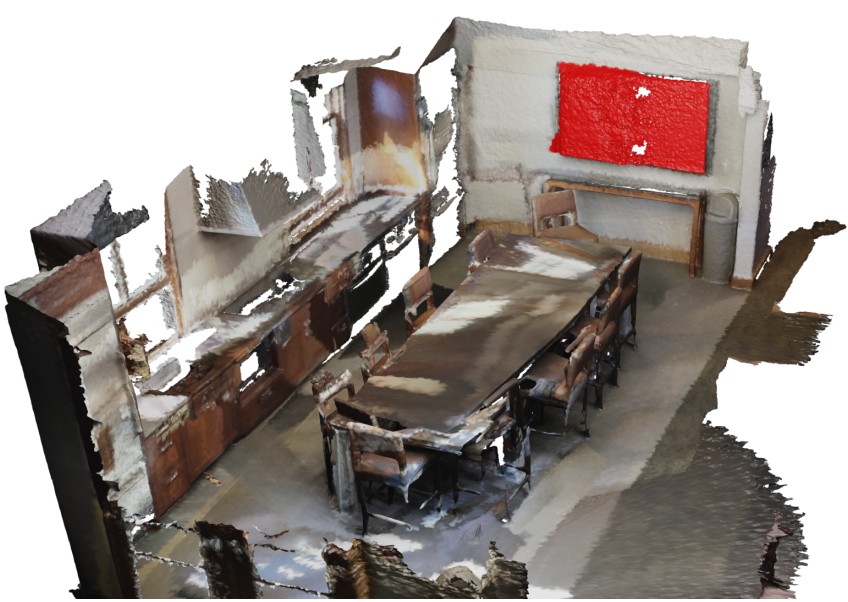

Figure 11: **Additional qualitative results on ScanNet200.** with text prompt *"Television"*.

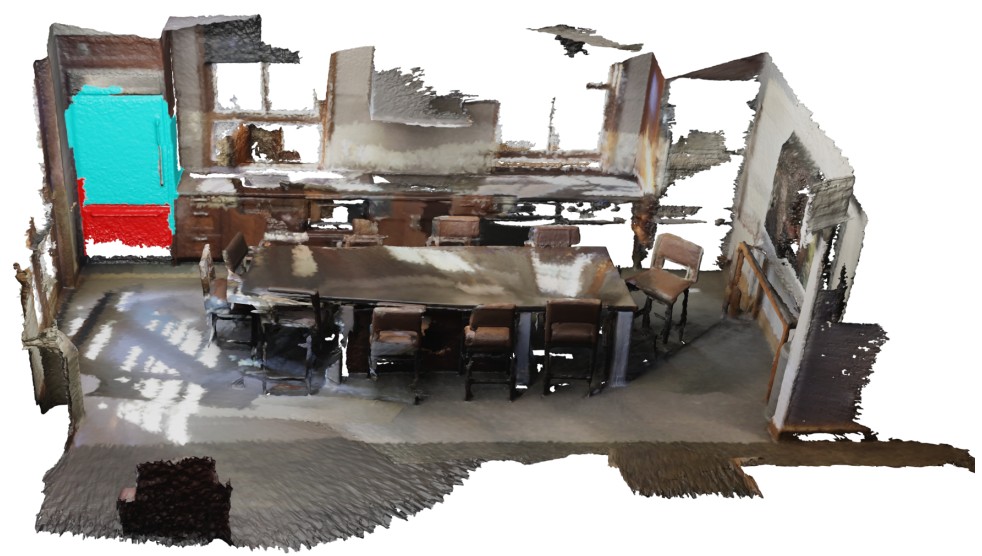

Figure 12: **Additional qualitative results on ScanNet200.** with text prompt *"Fridge"*.

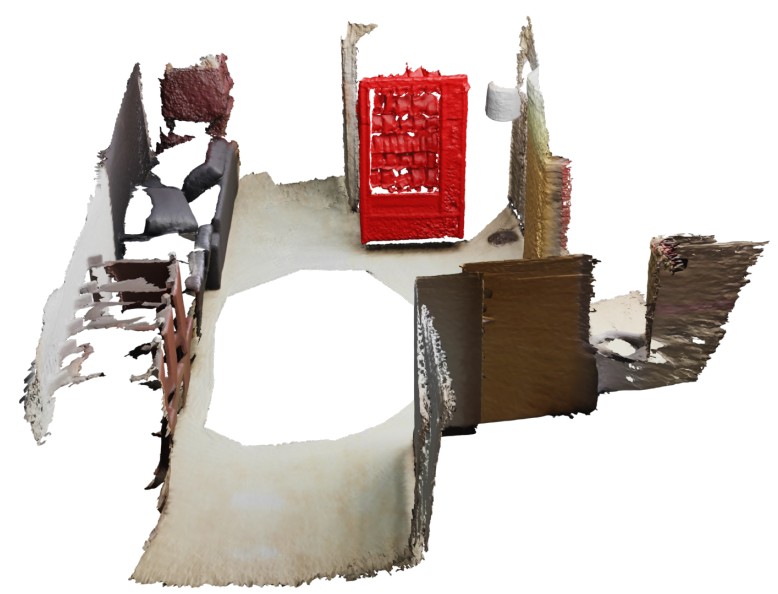

Figure 13: **Additional qualitative results on ScanNet200.** with text prompt *"Snacks"*.

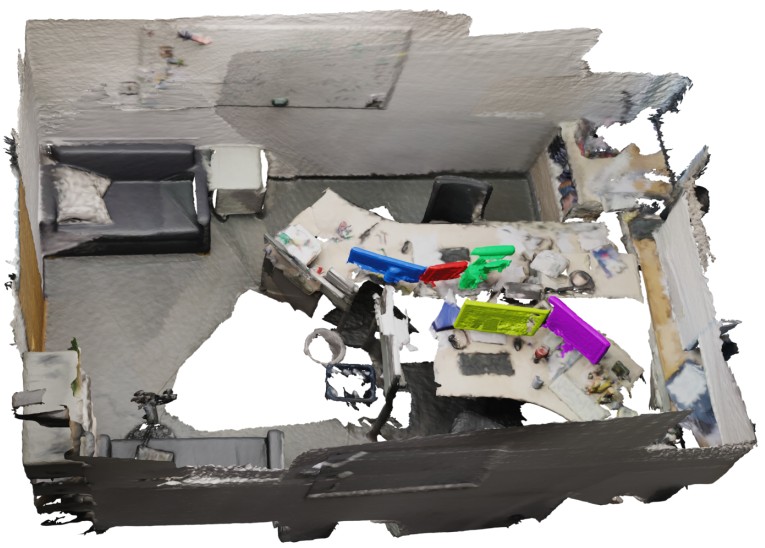

Figure 14: **Additional qualitative results on ScanNet200.** with text prompt *"Computer"*.

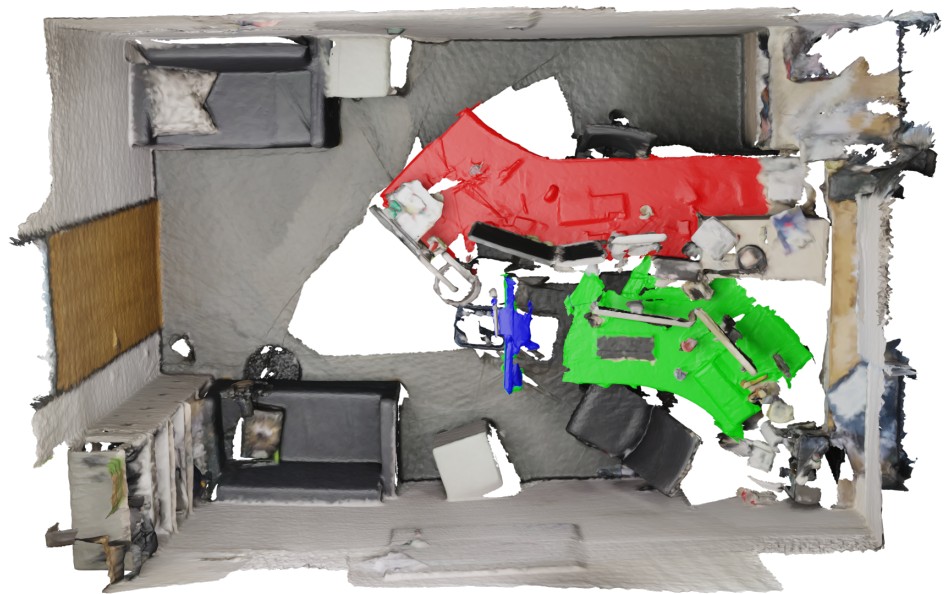

Figure 15: **Additional qualitative results on ScanNet200.** with text prompt *"Desk"*.

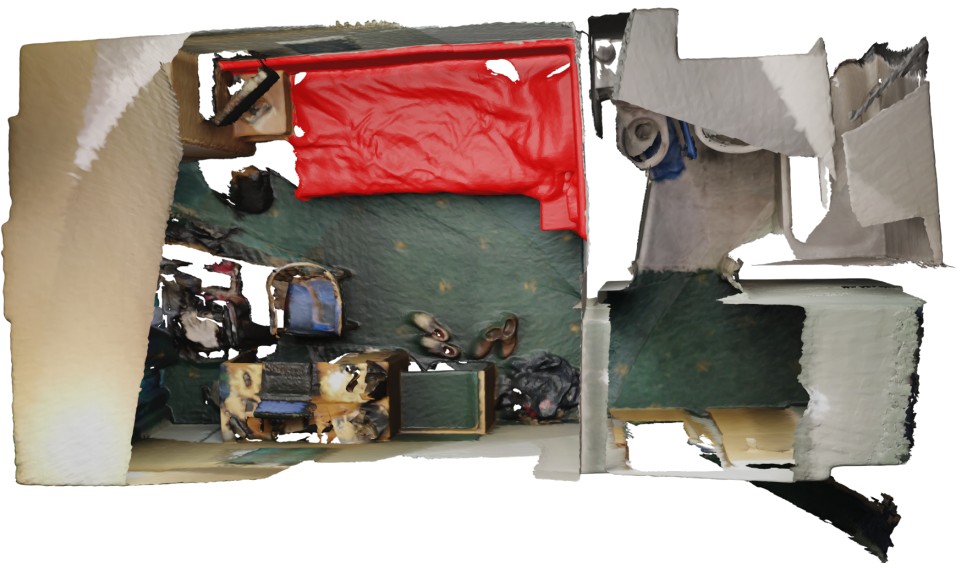

Figure 16: **Additional qualitative results on ScanNet200.** with text prompt *"Bed"*.

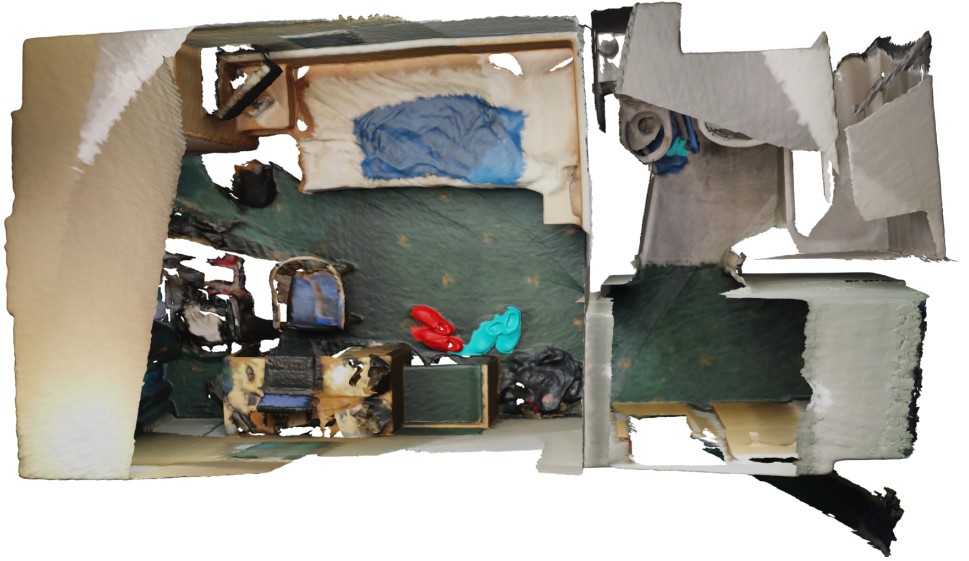

Figure 17: **Additional qualitative results on ScanNet200.** with text prompt *"Shoes"*.

## REFERENCES

Muhammad Muzammal Naseer, Kanchana Ranasinghe, Salman H Khan, Munawar Hayat, Fahad Shahbaz Khan, and Ming-Hsuan Yang. Intriguing properties of vision transformers. *Advances in Neural Information Processing Systems*, 34:23296–23308, 2021.