# OpenReview forum: "Open-YOLO 3D: Towards Fast and Accurate Open-Vocabulary 3D Instance Segmentation"
_ICLR.cc/2025/Conference — ICLR 2025 Oral_

### Official Review · Reviewer_rQL1 · 2024-11-02

**Soundness:** 3
**Presentation:** 4
**Contribution:** 3
**Rating:** 10
**Confidence:** 5

**Summary:**

This paper aims to deal with the open-vocabulary 3D instance segmentation task with a fast and cost-effective approach by utilizing a YOLO-style design. A Multi-View Prompt Distribution method is proposed to effectively fuse the multi-view information. The low granularity label maps are proposed to only use 2D detectors to predict prompt IDs for 3D instance masks. Experimental results demonstrate the state-of-the-art performance of the proposed method. The speed of the proposed method is about 16 times faster than that of existing methods.

**Strengths:**

1.	The proposed method is simple yet effective.
2.	The two proposed designs are helpful for 3D instance segmentation with meaningful designs.
3.	The experimental results show that the proposed method could achieve good performance while remaining very efficient.

**Weaknesses:**

1.	Is it possible to extend the proposed method on panoptic segmentation of 3D scenarios? Please present your design briefly for this.
2.	As shown in Table 1, the inference time of the proposed method is 21.8, which is slower than OpenScene (3D Distill). Please add the explanation for this phenomenon in the corresponding text (first paragraph of Section 5.1).
3.	In Line 405, it should be 4.29 but not 04.29 for OpenScene (3D Distill).

**Questions:**

Please draw my concerns listed in the Weaknesses part.

---

> ### Author Response · Authors · 2024-11-20
>
> We thank the reviewer for the encouraging and constructive comments; please find below our response to the mentioned questions.
>
> ### Q1: Is it possible to extend the proposed method on panoptic segmentation of 3D scenarios? Please present your design briefly for this.
>
> To perform Open Vocabulary panoptic segmentation, our pipeline can be used as is with background object class names in the input class list (walls, floor, ceiling), since YoloWorld can also generate bounding boxes for these categories. A panoptic segmentation network can be used for the proposal generation network, such as OneFormer3D [1].
>
> ### Q2: As shown in Table 1, the inference time of the proposed method is 21.8, which is slower than OpenScene (3D Distill). Please add the explanation for this phenomenon in the corresponding text (first paragraph of Section 5.1).
>
> In our revised version, we will provide a detailed explanation of this result. This is primarily because OpenScene relies on a U-Net model to predict CLIP features for each 3D point. This U-Net is trained by minimizing the cosine similarity loss between the predicted 3D CLIP features and the CLIP features aggregated from 2D; the 2D features are extracted through projections of the 3D point cloud on the predictions from 2D open-vocabulary semantic segmentation networks and then aggregated into per-3D-point CLIP features.
>
> - **OpenScene 3D Distill:** This approach is much faster since it relies on per-point CLIP features predicted using the trained 3D U-Net only, which takes less than a second to predict.
> - **OpenScene 2D:** Bypasses the 3D backbone predictions, directly constructing per-3D-point CLIP features from 2D open-vocabulary semantic segmentation. However, this method is significantly slower, requiring inference from many multi-view frames to generate the 3D features.
>
> ### Q3: In Line 405, it should be 4.29 but not 04.29 for OpenScene (3D Distill).
>
> We thank the reviewer for highlighting the typo; we will correct it and update it in the revised version.
>
> [1] Kolodiazhnyi, Maxim, et al. "Oneformer3d: One transformer for unified point cloud segmentation." CVPR 2024.

---

> > ### Comment · Reviewer_rQL1 · 2024-11-26
> >
> > Thanks for your quick and insightful reply!
> > After viewing all the review comments of all reviewers, I believe my concerns have been well addressed and thus keep my rating.

---

### Official Review · Reviewer_C5qy · 2024-11-04

**Soundness:** 3
**Presentation:** 3
**Contribution:** 3
**Rating:** 8
**Confidence:** 4

**Summary:**

The authors propose an efficient method for open-vocabulary 3D instance segmentation to enhance the real-time capability. Unlike existing methods that rely on obtaining 2D masks and category labels from 2D foundation models (like SAM and CLIP), the authors propose a novel approach, Open-YOLO3D, which only leverages bounding boxes generated by 2D object detectors. Moreover, the authors propose a Multi-View Prompt Distribution (MVPDist) method to endeavor promising performance in recognition. The experimental results demonstrate the promising real-time performance of the method proposed by the authors.

**Strengths:**

1.	The authors only utilize the bounding boxes from 2D object detectors to alleviate the redundancy brought by 2D masks, which demonstrates a significant improvement in inference speed compared to OpenMask3D.
2.	The authors propose a Multi-View Prompt Distribution to obtain reliable category labels form 3D masks, the experimental results evaluation on the ScanNet200 and Replica datasets prove the efficiency of the method proposed by the authors.
3.	The paper is well-structured, and the connection between the proposed method and the motivation is coincident.

**Weaknesses:**

1.	The improvements of the segmentation performance observed in Open-YOLO3D primarily arise from the enhanced category recognition, which is likely from the prior knowledge of the pre-trained YoloWorld model.
2.	The challenges inherent in Open-YOLO3D closely resemble those faced by Open3DIS, as both methods rely on pre-trained models for generating 3D proposals. As discussed in Open3DIS, the pre-trained 3D models have limited capabilities when it comes to detecting uncommon categories. The representation of 3D data for open vocabulary instance segmentation might be uncultivated and limited.
3.	Recent studies [1] have indicated that OpenMask3D performs poorly on certain outdoor datasets, such as NuScenes. Does Open-YOLO3D face similar challenges in effectively identifying sparse-diverse and less common categories in outdoor environments?

[1] Open-Vocabulary SAM3D: Towards Training-free Open-Vocabulary 3D Scene Understanding.

**Questions:**

Please see the weakness.

---

> ### Author Response · Authors · 2024-11-20
>
> We sincerely thank the reviewer for their thoughtful feedback, which will greatly contribute to improving the clarity and quality of our work. Please find our detailed responses below.
>
> ## Q1: On the Improvements of the Segmentation Performance Observed in Open-YOLO3D
>
> Even though Open-YOLO3D's enhanced segmentation performance can also be attributed to the high performance of the YoloWorld model, we argue that a naive 3D mask classification method cannot achieve the best out of 2D object detectors. To further demonstrate this, we conducted several experiments, with the results summarized in the table below.
>
> In row 0, we match 2D bounding boxes with the highest confidence scores from YoloWorld to 3D masks based on the IoU overlap of the 3D mask projections. The class prediction of the best-matching bounding box was then assigned to the corresponding 3D mask.
>
> In row 1, YoloWorld was used to generate bounding boxes, which were then leveraged to create multi-view crops. These crops were processed to predict clip features, which were subsequently aggregated into a single feature representing a 3D mask.
>
> Both approaches yielded significantly poorer results compared to our proposed MVDist method. This suggests that relying solely on the prior knowledge of pre-trained models is insufficient for effectively utilizing multi-view predictions from object detectors. In contrast, our approach leverages MVPDist, which encodes point frequency across multiview frames and integrates class predictions projected onto LG maps. This method achieves consistently superior performance with minimal computational overhead, relying only on selection operations.
>
> | Row ID | Method                                               | mAP  |
> |--------|------------------------------------------------------|------|
> | 0      | Box w/ highest confidence + IoU (YoloWorld classes) | 19.9 |
> | 1      | Crops from Box w/ highest confidence + IoU (CLIP features) | 32.5 |
> | 2      | Crops from SAM (OpenMask3D codebase)                | 33.0 |
> | 3      | **MVPDist + LG maps (ours)**                            | **46.2**|
>
> ## Q2: On the inherent challenges in Open-YOLO3D and their resemblance to those faced by Open3DIS
>
> Our 3D mask classification method can be used with any class-agnostic mask generation pipeline. Table 7 in our paper presents the results of using the Open3DIS approach to generate 3D instance masks by clustering 2D masks obtained from SAM, which were prompted with bounding boxes provided by an object detector. Additionally, we append the 3D masks generated by a 3D proposal network similar to Open3DIS; we use ISBNET similar to Open3DIS instead of Mask3D for a fair comparison with Open3DIS.
>
> While SAM's slower processing adds an extra 4.5 minutes to generate 3D proposals from 2D masks, this method achieves improved performance in terms of mAP compared to Open3DIS, while still being 1 minute faster overall.
>
>
> ## Q3: On the effectiveness in identifying sparse-diverse and less common categories in outdoor environments
> We conducted an experiment to evaluate our method on the NuScenes dataset using Mask3D as the proposal network, which was trained on the ScanNet200 dataset. The mean average precision (mAP) of all models is reported in the table below, with comparisons made against OpenMask3D and SAM3D. We report the results of OpenMask3D and SAM3D as in the SAM3D paper.
>
> For evaluation, since NuScenes provides 3D bounding boxes, we generate instance ground truth masks by masking the points within each box. For the input point cloud to Mask3D, we use the LiDAR point cloud represented in the global coordinate system. Each point is assigned an RGB color by projecting it onto the camera images using the intrinsic and extrinsic parameters of each camera.
>
> The results highlight challenges in generalizing to outdoor environments from models pre-trained on indoor datasets like ScanNet200, primarily due to LiDAR data sparsity and limited multi-view frames. Both Mask3D and OpenMask3D struggle to adapt to outdoor scenes due to low-quality 3D proposals from Mask3D, pre-trained on ScanNet200. However, they outperform SAM3D in indoor scenes, achieving superior results on ScanNet200.
>
> On NuScenes, the method performs comparably to OpenMask3D, with performance constrained by the dataset’s limited multi-view frames, as most instances are associated with only a single 2D frame.
>
> || 3D  Pretraining | 2D IS Network | NuScenes | ScanNet200 |
> |-------------------------|-------------------------|---------------|----------|------------|
> | OpenMask3D             | ScanNet200    | None     | 0.5        | 15.4       |
> | SAM3D                  | None          | SAM      | 8.9        | 9.0        |
> | Open3DIS               | None          | SAM      | -          | 18.6       |
> | Open3DIS               | ScanNet200    | SAM      | -          | 23.7       |
> | **Ours**                  | ScanNet200    | None     | 0.52       | 24.7       |

---

### Official Review · Reviewer_4bgM · 2024-11-04

**Soundness:** 4
**Presentation:** 4
**Contribution:** 3
**Rating:** 8
**Confidence:** 3

**Summary:**

This paper proposes Open-YOLO 3D, which is an open-vocabulary 3D instance segmentation framework that efficiently combines 2D object detection and 3D mask generation. The key idea of this paper is its reliance on bounding box predictions from a 2D open-vocabulary object detector and the subsequent use of these predictions for efficient 3D mask proposal and labeling. Unlike prior methods that use computationally intensive models such as SAM and CLIP for feature lifting from 2D to 3D, this paper uses a novel Multi-View Prompt Distribution (MVPDist) and Accelerated Visibility Computation (VAcc) methods to speed up the segmentation process. The framework this paper proposed achieves up to 16x faster inference while keeping competitive or better accuracy.

**Strengths:**

This paper proposed a novel framework, which uses 2D object detection for 3D instance segmentation. The model they presented reduced computational overhead significantly. The Accelerated Visibility Computation (VAcc) leverages tensor operations and GPU batch processing, enabling highly parallelized visibility computation. This contributes to the following speed improvements without compromising performance. By integrating a high-performing 2D open-vocabulary detector, the framework retains strong zero-shot performance, which is important for real-world applications that use new or unknown object types.

It also includes detailed experiments that showcase Open-YOLO 3D's speech and accuracy, and highlight its performance above state-of-the-art approaches like Open3DIS and OpenMask3D. The paper also includes comprehensive ablation studies to demonstrate the improvement of each component.

The overall writing is clear and the framework will be beneficial for related research.

**Weaknesses:**

I like the overall framework this paper presents and appreciate its contribution to 3D instance segmentation by introducing an inference-efficient model, but I still have some concerns about it:

While the paper mentions that VAcc uses tensor operations, a deeper explanation or complexity analysis comparing it to conventional iterative methods would strengthen the understanding of its true computational advantage, and the reason why it can achieve faster inference speed. I believe the paper clearly demonstrates the operation of this proposed algorithm, however, more explanation about why it is efficient and how much computation cost it saves will better demonstrate the paper's contribution.

The method relies on the quality of the 2D object detector, and this might be an issue if the 2D views are suboptimal (for example poor lighting, and occlusions). A more extensive analysis or discussion on how 2D detection failures propagate through the pipeline would add value.

**Questions:**

Could you provide a more detailed theoretical analysis or complexity comparison of VAcc with the conventional method?
How does the method perform when the 2D object detector encounters difficult conditions, such as poor lighting or significant occlusion, if there's any evaluation of the robustness under such conditions?
What are the potential strategies for mitigating errors from misclassifications made by the 2D object detector, and how do they affect the 3D mask assignments?
I'm also particularly interested in the discussion in your limitation section, I was wondering would integrating fast 2D segmentation models, as mentioned, be feasible within your current framework? How might this affect both performance and speed?

---

> ### Author Response · Authors · 2024-11-20
>
> We thank the reviewer for their constructive comments, which will significantly enhance the clarity of our work. Kindly find below our detailed responses.
>
> ## Q1: Complexity Analysis of VAcc
>
> The visibility computation method in **OpenMask3D** projects 3D points onto frames to identify visible points, using depth maps to filter out occluded points and ensure points fall within the frame's dimensions. Visibility is determined by checking if a point's projected depth matches the depth map value. This process iterates through N 3D masks, M frames, and P points, resulting in both computational and time complexities of O(N × M × P). However, it uses sequential loops, limiting parallelization efficiency.
>
> In our proposed method, **VAcc**, we reformulate the visibility computation using tensor operations, enabling parallelization across multiple cores. While maintaining the same computational complexity O(N × M × P), it reduces time complexity to O(N × M × P / c), where c is the number of cores. This makes VAcc a faster alternative to the method used in OpenMask3D.
>
> ## Q2: Effect of Occlusion and Light Conditions
>
> ### Occlusion Analysis
>
> This analysis evaluates model robustness to occlusion using the testing protocol from [1]. Images are systematically occluded by dropping patches (1% of image size) centered on pixels from 2D projections of 3D ground truth instance masks. The experiments include:
>
> **1. Salient Patch Drop**
> - Simulates occlusion of semantically important foreground objects by removing patches from 2D masks derived from instance masks, excluding walls, floors, and ceilings.
> - Larger masks have more patches removed to uniformly test the impact of occlusion on detector performance.
>
> **2. Non-Salient Patch Drop**
> - Focuses on background occlusion by removing patches from areas corresponding to walls, floors, and ceilings.
> - Tests the detector's robustness to non-critical occlusion.
>
> These experiments assess how meaningful and arbitrary occlusions affect object detection under extreme conditions.
>
> Table: Occlusion analysis
> | Percentage  | 0   | 5   | 30  | 55  | 80  | 95  | 100  |
> |-------------------------------|-----|-----|-----|-----|-----|-----|------|
> | Salient           | 46.2| 40.6| 18.1| 8.0 | 7.0 | 0.5 | 0.0  |
> | Non-Salient       | 46.2| 44.5| 39.4| 39.7| 38.0| 36.1| 35.3 |
>
>
> We show that our method is robust to moderate levels of salient occlusion where the important objects are occluded while being extremely robust when the non-salient objects are occluded.
>
> ### Analysis of Extreme Lighting Conditions:
>
> We test the model across different levels of lighting conditions, starting with 0.05 intensity (almost black) up to 2.5 intensity (extremely bright). A brightness level of 1 corresponds to the original lighting conditions. The robustness results demonstrate that the pipeline performs well under moderate fluctuations in light intensity but encounters challenges in extremely low-light conditions.
>
> Table: Analysis under different Lighting Conditions
> | Brightness  | 0.05 | 0.25 | 0.5  | 0.75 | 1    | 1.25 | 1.5  | 1.75 | 2    | 2.5|
> |-------------|------|------|------|------|------|------|------|------|------|------|
> | mAP (ours)  | 15.6 | 39.3 | 40.8 | 44.7 | 46.2 | 43.0 | 41.1 | 40.0 | 39.5 | 35.6 |
>
>
> ## Q3: How MVPDist Mitigates Misclassifications from Object Detectors
>
> We observe that object detectors sometimes predict bounding boxes for incorrect classes with high confidence. Consequently, a naive 3D mask labeling approach that associates a 3D mask with the bounding box of highest confidence across views can lead to incorrect predictions. To further demonstrate this, we conduct the experiment in row in the table below:
>
> | Row ID | Methodology                                              | mAP  |
> |--------|----------------------------------------------------------|------|
> | 0      | Box w/ highest confidence + IoU (YoloWorld classes)      | 19.9 |
> | 1      | Crops from Box w/ highest confidence + IoU (CLIP features)| 32.5 |
> | 2      | Crops from SAM (OpenMask3D codebase)                     | 33.0 |
> | 3      | **MVPDist + LG maps (ours)**                                 | 46.2 |
>
>
> Our MVPDist method significantly outperforms confidence- and IoU-based 3D-to-2D instance matching by encoding 3D instance information into the distribution rather than relying solely on confidence scores or IoU from YoloWorld.
>
> ## Q4: How Does Integrating 2D Segmentation Models Affect Speed and Performance?
>
> Table 7 in our paper presents the results of using the Open3DIS approach to generate 3D instance masks by clustering 2D masks obtained from SAM, which were prompted with bounding boxes provided by an object detector. While SAM's slower processing adds an extra 4.8 minutes to generate 3D proposals from 2D masks, this method achieves improved performance in terms of mAP compared to Open3DIS, while still being ~1 minute faster overall.
>
> [1]Naseer, et al. "Intriguing properties of vision transformers." NeurIPS 2021

---

> > ### Comment · Reviewer_4bgM · 2024-11-26
> >
> > Thank you for the clear and thorough responses! The reformulation of visibility computation with tensor operations and the detailed robustness analyses under occlusion and lighting conditions are clear to me now, and I suppose they are also well supported by quantitative results. The effectiveness of MVPDist in mitigating misclassifications is also clearly demonstrated, and the integration of 2D segmentation models balances performance improvements with reasonable processing times. Your clarifications and additional experiments have addressed my concerns, and I will update my score accordingly. I appreciate your excellent work!

---

### Official Review · Reviewer_LA4L · 2024-11-06

**Soundness:** 2
**Presentation:** 3
**Contribution:** 2
**Rating:** 5
**Confidence:** 4

**Summary:**

This paper primarily aims to achieve faster open-vocabulary 3D instance segmentation compared with existing methods like OpenMask3D. To realize this target, this work first uses a 3D instance segmentation network to generate segmentation proposals. Then, the output of an open-vocabulary 2D object detector as well as some designed 3D information is employed to derive the categories of these proposals.

**Strengths:**

1. **[Efficiency]** The experimental results suggest that the proposed method achieves high precision with a significantly better speed compared with most methods, and efficiency is important for practical deployment.

2. **[Clearness]** This paper explains its main contribution, how to assign class predictions to 3D proposals, with great clarity. The implementation details are elaborated sufficiently.

**Weaknesses:**

1. **[Insufficient Academic Contributions]**: This work just combines the output of a 3D segmentation network and a well-implemented open-vocabulary 2D object detector to realize open-vocabulary 3D object detection (similar to existing open-world segmentation method, just with a replacement of the post network to 2D object detector), which is trivial. It is much faster than previous methods because previous methods are developed based on models like SAM and CLIP. This work employs more efficient and suitable existing models. Therefore, although this work is sound in terms of engineering, its real academic contribution and new insights are plain.

2. **[Insufficient Ablation Study]** As the method is efficient because it makes good use of existing models, it is important to clearly analyze how these models contribute to the efficiency, which will guide future works on how to develop an efficient open-vocabulary pipeline. However, this work fails to do so.

3. **[Misleading title]** The method name OPEN-YOLO 3D seems to be unsuitable. YOLO is a 2D object detector while the task is about 3D point cloud segmentation. Although the method utilizes the output of YOLO-World to generate class predictions, the method name is still a little misleading.

**Questions:**

See weakness.

---

> ### Author Response · Authors · 2024-11-20
>
> We thank the reviewer for the constructive feedback. A well-documented code along with pre-trained models will be publicly released. Our detailed responses are provided below.
>
> ## W1: On the Contributions
>
> Efficiently labeling 3D masks using predictions from object detectors is a challenging task. In our work, we propose **LG+MVPDist**, a method that enables accurate class predictions from multiview images, effectively addressing the limitations of 2D object detectors. Unlike CLIP features, which can be aggregated into a single strong feature that encodes multiview information about the object, object detectors generate a set of bounding boxes for all input classes, making it difficult to efficiently match the best 2D box to its corresponding 3D mask.
>
> In the experiment (row 0) detailed in **Table I**  below, the class label for the 3D mask is determined by selecting the YOLOWorld-predicted bounding box with the highest IoU overlap with the 2D bounding box constructed by projecting the 3D mask onto the corresponding view. The class prediction is then taken from the bounding box with the highest confidence score across the views with top-k visibility.
>
> In **row 1**, the matched bounding box is further used to generate crops for constructing visual CLIP features for the 3D masks, following a similar approach to OpenMask3D. These visual features are then used to predict the class label.
>
> Our results highlight that **LG Maps** and **MVPDist** significantly outperform both of these techniques. This improvement is primarily due to the limitations of object detectors, which can sometimes assign incorrect class labels with high confidence.
>
> **MVPDist** mitigates this issue by filtering out incorrect labels from certain views. It achieves this by constructing class distributions from multiple views and assigning the mask the most frequent class. Since MVPDist encodes the frequency of points projected onto different views, the distribution leans toward views with higher point densities. This ensures that the classification process prioritizes frames where the object is most clearly represented, reducing the impact of less informative views and improving overall accuracy.
>
> We hope our simple and effective approach will serve as a solid baseline and help ease future research in fast and accurate open-vocabulary 3D instance-level recognition.
>
> ### Table I: Contributions Analysis
>
> | Row ID | Methodology                                       | mAP  |
> |--------|--------------------------------------------------|------|
> | 0      | Box w/ highest confidence + IoU (YoloWorld classes) | 19.9 |
> | 1      | Box w/ highest confidence + IoU (CLIP features)    | 32.5 |
> | 2      | Crops from SAM (OpenMask3D codebase)              | 33.0 |
> | 3      | MVPDist + LG Maps                                 | 46.2 |
>
>
> ## W2: On Additional Ablation
>
> We appreciate the reviewer's feedback and will restructure the ablation table for better clarity in the revised version. Below, we present our updated ablation study, introducing an experiment in **row 0** where we don’t use our proposed MVPDist+LG maps. In this setup, we assign the 3D mask a class label based on the YoloWorld prediction with the highest IoU overlap and confidence across the views. This experiment in row 0 relies solely on YoloWorld predictions.
>
> Our results demonstrate that incorporating **MVPDist+LG maps** significantly enhances performance by leveraging point frequency across views to predict class labels, rather than depending entirely on the 2D object detector, which can occasionally assign incorrect class labels. Our observations show that YoloWorld occasionally predicts the wrong class with the highest confidence across views for the same 3D instance.
>
> ### Updated Ablation Table
>
> | Row ID | Deducted Components                              | mAP  | Time (s)  |
> |--------|-------------------------------------------------|-------|-----------|
> | 0      | Ours - MVPDist - Vacc (w/ YoloWorld classes)    | 19.9  | 392.02    |
> | 1      | Ours - MVPDist - Vacc (w/ CLIP features)        | 32.5  | 396.89    |
> | 2      | Ours - Vacc                                     | 46.2  | 376.42    |
> | 3      | Ours                                            | 46.2  | 17.86     |
>
>
> ## W3: Title
>
> We note that the motivation behind the title is to highlight our contributions towards effectively adapting the popular YOLO-based architecture for real-time open-vocabulary 3D instance segmentation. While YOLO-based design has been recently explored for 2D open-vocabulary detection literature, we are the first to investigate and adapt it for 3D open-vocabulary instance segmentation.

---

### Official Review · Reviewer_qcNj · 2024-11-08

**Soundness:** 3
**Presentation:** 3
**Contribution:** 3
**Rating:** 8
**Confidence:** 3

**Summary:**

This paper introduces an efficient 3D mask labeling method that leverages multi-view 2D label maps, referred to as Low Granularity (LG) Label Maps, created from 2D object bounding boxes to label 3D instances. The 3D instance (mask) proposals are generated using a pre-trained class-agnostic 3D segmentation method. To address object occlusion across different viewpoints, an Accelerated Visibility Computation (VACC) method is introduced, enabling rapid calculation of visibility matrices using intrinsic and extrinsic parameters.

**Strengths:**

1. The paper is well-organized, and the ideas are clearly illustrated.
2. This paper introduces a novel approach for efficient open-vocabulary 3D instance labeling by leveraging 2D bounding box priors from a fast 2D object detector, demonstrating superior performance and time efficiency in experimental results.
3. A fast visibility computation algorithm (VAcc) is proposed to accelerate the process of associating 2D label maps with 3D proposals that may be occluded in some views.  This algorithm demonstrates both efficiency and robustness to variations in label map granularity.

**Weaknesses:**

1. The foundation of the proposed method is built upon the class-agnostic 3D segmentation model, Mask3D, which is used to generate 3D mask proposals. However, this paper lacks sufficient evidence to demonstrate Mask3D's effectiveness and generalizability for open vocabulary instance proposals.
2. The experimental evaluation of the proposed method for open-vocabulary 3D instance segmentation is relatively limited (only Table 6).

**Questions:**

1. In Table 1, does the class-agnostic Mask3D model have access to mask annotations for the same classes as those in the validation set? Do the other methods use the same class-agnostic segmentation model?

2. Since the proposed approach relies on a class-agnostic 3D instance generation model, what are the advantages of using only mask annotations, rather than both instance and label annotations, for training? I mean, are there practical scenarios where only mask annotations are available?

3. What does the tag "(Closed Vocab)" mean in Table 1? Does it indicate that the Mask3D method uses both mask annotations and object class annotations for training?

4. What is the performance of Mask3D (Closed Vocab.) on the Replica dataset?

Minor：

1. In line 092, a comma is missing after "multi-view information".

---

> ### Author Response · Authors · 2024-11-20
>
> We sincerely appreciate the reviewer’s thoughtful and valuable feedback. A well-documented code along with pre-trained models will be publicly released. Our detailed responses are provided below.
>
> ## W1: Regarding the effectiveness and generalizability of baseline Mask3D for open vocabulary instance proposals
>
> We conducted experiments to evaluate Mask3D’s generalizability to unseen geometries and assess its ability in terms of mask proposal generation. The results are presented in **Table 2** and **Table 6** (main manuscript). Furthermore, we report in Table below the results of Table 2 on Novel/Base split of replica classes; the split will be publicly released. The Base classes (39 classes) consist of those that are semantically similar to at least one class from the ScanNet200 dataset, while the Novel classes (9 classes) include all remaining ones. Furthermore, we highlight that our method can be used with any class-agnostic proposal generation method. We report in **Table 7** of our manuscript that our approach (**MVPDist + LG maps**), which utilizes 3D proposals clustered from 2D masks following the Open3DIS methodology, achieves superior results compared to Open3DIS (point-wise CLIP features).
>
> - **Table 2**: Mask3D was trained on ScanNet200 and tested on Replica. This demonstrates Mask3D’s ability to generate proposals for out-of-distribution (OOD) datasets where objects exhibit distinct geometries and characteristics. Furthermore, we show in the table below that our method performs much better compared to Open3DIS on the Replica Base/Novel split.
>
> - **Table 6**: Mask3D was trained on ScanNet’s 20 classes and tested on ScanNet200, which includes 200 classes. Out of these, 53 classes are considered base classes (which are semantically similar to the 20 classes in ScanNet), while the remaining 147 classes are labeled as novel. This indicates that Mask3D encountered objects with entirely different geometries during testing, showcasing its robustness in handling novel objects in indoor environments. We adopted the split proposed by OpenMask3D authors.
>
> Table: results on Base/Novel replica splits with Mask3D as proposal network
> | Method      | mAP  | mAP_base | mAP_novel |
> |-------------|------|----------|-----------|
> | Open3DIS    | 18.2 | 19.9     | 10.8      |
> | **Ours**    | 23.7 | 26.4     | 11.9      |
>
> ## W2: On the experimental evaluation of the proposed method for open-vocabulary 3D instance segmentation
>
> We use the same evaluation setting that OpenMas3D and Open3DIS adopt. For proposal generation, OpenMask3D uses Mask3D trained on the ScanNet200 training set, whereas Open3DIS uses ISBNET, also trained on the ScanNet200 training set.
>
> 1. **In-Distribution Test (Table 1)** : Evaluation on ScanNet200 validation set with a proposal network trained on ScanNet200 training set. We compare against Open3DIS and OpenMask3D under the same setting.
>
> 2. **Indoor Out-of-Distribution Test (Table 2)**: With proposal network trained on ScanNet200 and tested on the Replica dataset to assess generalizability to unseen indoor distributions.
>
> 3. **Generalizability to Novel Geometries (Table 6)**: Trained on ScanNet with 20 classes and tested on ScanNet200, to evaluate adaptation to new geometries and categories.
>
> ## Q1: In Table 1, does the class-agnostic Mask3D model have access to mask annotations for the same classes as those in the validation set? Do the other methods use the same class-agnostic segmentation model?
> Yes, in this evaluation setting, the classes used for training are similar to those used for validation. OpenMask3D utilizes Mask3D, trained on the ScanNet200 training set (the same pre-trained network in ours), while Open3DIS uses ISBNET, also trained on the ScanNet200 training set. According to the Open3DIS paper (page 8, Table 9), both ISBNET and Mask3D deliver comparable results.
>
> ## Q2: Since the proposed approach relies on a class-agnostic 3D instance generation model, what are the advantages of using only mask annotations, rather than both instance and label annotations, for training? Are there practical scenarios where only mask annotations are available?
> In all previous methods (OpenMask3D, Open3DIS), and ours, both class and mask annotations are used during training. However, during inference, class predictions are disregarded and replaced with an open-vocabulary classification approach.
>
> ## Q3: What does the tag "(Closed Vocab)" mean in Table 1? Does it indicate that the Mask3D method uses both mask annotations and object class annotations for training?
> A closed-vocabulary tag indicates that the model was fully supervised on classes similar to those present in the validation set.
>
> ## Q4: What is the performance of Mask3D (Closed Vocab.) on the Replica dataset?
> Since the Replica dataset does not include a training set, it was not feasible to evaluate Mask3D in a fully supervised setting on this dataset.
>
> ## Q5: Typo on line 092
> We thank the reviewer and will fix the typo in the final version.

---

> > ### Comment · Reviewer_qcNj · 2024-11-27
> >
> > Thank you for your hard work and prompt response. While I still don’t understand how Mask3D achieves open vocabulary 3D instance proposals, particularly given its reliance on category information during training, I acknowledge that this issue was also not addressed in prior work. Therefore, I’ve decided to update my rating.

---

### Author Response · Authors · 2024-11-20

We thank all the reviewers (qcNj, LA4L, 4bgM, C5qy, rQL1 ) for their valuable feedback and constructive comments, which will undoubtedly enhance the clarity and overall quality of our work. A well-documented code with the pre-trained models will be publicly released. [qcNj]This paper introduces a novel approach for efficient open-vocabulary 3D instance labeling. [LA4L] The experimental results suggest that the proposed method achieves high precision with a significantly better speed compared with most methods, [4bgM] The overall writing is clear and the framework will be beneficial for related research. [C5qy] the connection between the proposed method and the motivation is coincident. [rQL1]The experimental results show that the proposed method could achieve good performance while remaining very efficient.

As requested by the reviewers, we uploaded a new revised version by including the following in the manuscript (the updated sections are highlighted in blue in the revised version):
- We organized and updated the ablation study in Table 7, line 469.
- We fixed the typo in line 092.
- We fixed the typo in Table 405.
- We added an explanation regarding the performance of Openscene 3D distill in line 413.
- We added a theoretical complexity analysis in the supplementary material from line 11 to line 25.
- We added robustness study and analysis in supplementary material from line 25 to line 61.
-  We provided additional details regarding the generalization of the proposal network from ScanNet200 to the Replica base/novel split in the supplementary material (Table 3).

---

### Meta-Review · Area_Chair_kiqG · 2024-12-20

**Metareview:**

This paper presents an efficient approach to open-vocabulary 3D instance segmentation by leveraging 2D bounding box priors from a pre-trained open-vocabulary 2D object detector. The authors propose the Multi-View Prompt Distribution (MVPDist) method, which effectively utilizes multi-view information while addressing potential misclassification from the 2D object detector to generate reliable 3D instance masks. The experimental results demonstrate the promising real-time performance of the method.

Initially, the reviewers raised several concerns, including:

- The generalizability of Mask3D (qcNj)
- Insufficient evaluation (qcNj, LA4L)
- The engineering focus of the work with insufficient academic contribution (LA4L)
- Lack of explanation and complexity analysis of VAcc (4bgM)
- The robustness of the 2D object detector in extreme conditions (4bgM)
- Performance improvements primarily attributed to the pre-trained YoloWorld model (C5qy)
- Limited capabilities on uncommon categories (C5qy)
- Performance on outdoor datasets (C5qy)
- Feasibility of extending the method to 3D panoptic segmentation (rQL1)
- Slower inference times (rQL1)

The authors provided detailed responses to these concerns, and after the rebuttal, four reviewers (4bgM, rQL1, C5qy, qcNj) voted to accept the paper, while reviewer LA4L did not update their initial rating, which was borderline rejection. The AC noted that reviewer LA4L did not follow up during the author response period.

Upon reviewing the authors’ rebuttal to reviewer LA4L’s comments, the AC finds the authors’ responses to the second and third concerns - regarding the engineering focus and the academic contribution - particularly convincing. The first concern, regarding the academic contribution, is more subjective, but the AC agrees that the proposed Open-YOLO 3D method is effective for the open-vocabulary 3D instance segmentation task.

Given the resolution of most concerns and the strong experimental results, the AC recommends accepting this paper and believes it is appropriate to let the community assess its academic contribution.

**Additional Comments On Reviewer Discussion:**

Initially, the reviewers raised several concerns, including:

- The generalizability of Mask3D (qcNj)
- Insufficient evaluation (qcNj, LA4L)
- The engineering focus of the work with insufficient academic contribution (LA4L)
- Lack of explanation and complexity analysis of VAcc (4bgM)
- The robustness of the 2D object detector in extreme conditions (4bgM)
- Performance improvements primarily attributed to the pre-trained YoloWorld model (C5qy)
- Limited capabilities on uncommon categories (C5qy)
- Performance on outdoor datasets (C5qy)
- Feasibility of extending the method to 3D panoptic segmentation (rQL1)
- Slower inference times (rQL1)

The authors provided detailed responses to these concerns, and after the rebuttal, four reviewers (4bgM, rQL1, C5qy, qcNj) voted to accept the paper, while reviewer LA4L did not update their initial rating, which was borderline rejection. The AC noted that reviewer LA4L did not follow up during the author response period or the AC-reviewer discussion.

Upon reviewing the authors’ rebuttal to reviewer LA4L’s comments, the AC finds the authors’ responses to the second and third concerns - regarding the engineering focus and the academic contribution - particularly convincing. The first concern, regarding the academic contribution, is more subjective, but the AC agrees that the proposed Open-YOLO 3D method is effective for the open-vocabulary 3D instance segmentation task.

Given the resolution of most concerns and the strong experimental results, the AC recommends accepting this paper.

---

### Decision · Program_Chairs · 2025-01-22

Accept (Oral)